



**Comparison of model and ground observations finds snowpack and blowing snow both contribute**
**to Arctic tropospheric reactive bromine**
William F. Swanson[1], Chris D. Holmes[2], William R. Simpson[1], Kaitlyn Confer[3], Louis Marelle[45], Jennie
L. Thomas[4], Lyatt Jaeglé[3], Becky Alexander[3], Shuting Zhai[3], Qianjie Chen[6], Xuan Wang[7], Tomás
Sherwen[89]
[1]Department of Chemistry and Biochemistry and Geophysical Institute, University of Alaska Fairbanks,
Fairbanks, Alaska
[2]Department of Earth, Ocean and Atmospheric Science, Florida State University, Tallahassee, Florida
[3]Department of Atmospheric Sciences, University of Washington, Seattle, Washington
[4]Institut des Géosciences de l'Environnement (IGE), Institut Polytechnique de Grenoble, Grenoble, France
[5]Laboratoire Atmosphères Observations Spatiales (LATMOS), Sorbonne Université, Paris, France
[6]Department of Civil and Environmental Engineering, Hong Kong Polytechnic University, Hong Kong,
China
[7]School of Energy and the Environment, City University of Hong Kong, Hong Kong, China
[8]National Centre for Atmospheric Science, University of York, York, UK.
[9]Department of Chemistry, University of York, York, United Kingdom
*Correspondence to:* William F. Swanson (wswanson3@alaska.edu)
**Abstract**
Reactive halogens play a prominent role in the atmospheric chemistry of the Arctic during
springtime. Field measurements and models studies suggest that halogens are emitted to the atmosphere
from snowpack and reactions on wind-blown snow. The relative importance of snowpack and blowing
snow sources is still debated, both at local scales and regionally throughout the Arctic.  To understand
implications of these halogen sources on a pan-Arctic scale, we simulate Arctic reactive bromine
chemistry in the atmospheric chemical transport model GEOS-Chem. Two mechanisms are included: 1) a
blowing snow sea salt aerosol formation mechanism and 2) a snowpack mechanism assuming uniform
molecular bromine production from all snow surfaces. We compare simulations including neither
mechanism, each mechanism individually, and both mechanisms to examine conditions where one
process may dominate or the mechanisms may interact.  We compare the models using these mechanisms
to observations of bromine monoxide (BrO) derived from multiple-axis differential optical absorption
spectroscopy (MAX-DOAS) instruments on O-Buoy platforms on the sea ice and at a coastal site in
Utqiaġvik, Alaska during spring 2015.  Model estimations of hourly and monthly average BrO are
improved by assuming a constant yield of 0.1% molecular bromine from all snowpack surfaces on ozone
deposition. The blowing snow mechanism increases BrO by providing more surface area for reactive





bromine recycling. The snowpack mechanism led to increased BrO across the Arctic Ocean with
maximum production in coastal regions, whereas the blowing snow mechanism increases BrO in specific
areas due to high surface windspeeds. Our uniform snowpack source has a greater impact on BrO mixing
ratios than the blowing snow source. Model results best replicate several features of BrO observations
during spring 2015 when using both mechanisms in conjunction, adding evidence that these mechanisms
are both active during the Arctic Spring. Extending our transport model throughout the entire year leads
to predictions of enhanced fall BrO that are not supported by observations.
**1. Introduction**
Simulating Arctic halogen chemistry is a persistent problem for global models because processes
appear to differ between the Arctic and middle latitudes (Parrella et al., 2012; Schmidt et al., 2016).
Space-based instruments observe large column densities of reactive bromine across swaths of the Arctic
Ocean during the Arctic spring (Chance, 1998; Richter et al., 1998; Wagner and Platt, 1998). Increased
levels of tropospheric reactive bromine are associated with ozone depletion events (Barrie et al., 1988;
Foster et al., 2001; Koo et al., 2012; Halfacre et al., 2014) as well as  oxidation of gaseous elemental
mercury (Schroeder et al., 1998; Nghiem, 2013; Moore et al., 2014).  Bromine radicals have been
observed to lead directly to ozone depletion and mercury oxidation (Wang et al., 2019a).  Deposition of
oxidized mercury to the snowpack can have deleterious effects on the health of Arctic humans and
animals (AMAP, 2011). Arctic reactive bromine chemistry impacts tropospheric oxidative chemistry but
is not typically accounted for in global models. Model studies have found that reactive halogen chemistry
can explain the oxidation of gaseous elemental mercury (Holmes et al., 2010) and reduce radiative forcing
from ozone (Sherwen et al., 2016c). Replicating reactive halogen chemistry in models requires inclusion
of multi-phase chemical reactions as well as mechanisms affecting sea salt aerosol particle production and
chemical reactions within the snowpack.
These increased levels of tropospheric reactive bromine radicals are a product of heterogeneous
photochemical reactions at the interface between air and saline surfaces such as surface snowpack and sea
salt aerosols (Saiz-Lopez and von Glasow, 2012; Simpson et al., 2015). Figure 1 depicts the gas-phase,
heterogeneous, and photochemical reactions thought to control tropospheric bromine, all of which are
included in the model and results presented in this manuscript. Bromine radicals (Br) are produced by
photolysis of molecular bromine (P1) and react with ozone to form bromine monoxide (BrO) (R2).
Under sunlit conditions, BrO is most often photolyzed back to Br radicals and an oxygen atom (P2) that
then most often reforms ozone, resulting in a null cycle. Due to this rapid interchange of Br and BrO,
these two compounds form the $BrO_x$ family. If processes other than BrO photolysis (P1) convert BrO
back to Br without producing ozone, the imbalance between these other processes and P1 result in net



ozone depletion. For example, ozone is depleted through R6 or R7 when BrO reacts with another halogen
oxide to form either $Br_2$ or BrCl, or through other more extended processes. A reactive halogen activating
cycle occurs when a BrO radical reacts with a hydroperoxy ($HO_2$) radical in R5 to form gaseous
hypobromous acid (HOBr). Heterogeneous chemistry can occur on a saline surface between HOBr and
particulate bromide (p-$Br^-$) in HR1 forming $Br_2$ or particle chloride (p-$Cl^-$) in HR6 forming BrCl.  For
each cycle of reactions P1, R2, R5, and HR1, one hydroperoxy radical is removed from the atmosphere,
one bromine radical is released to the atmosphere, and one ozone molecule is destroyed. This process of
activation of particulate bromide to $Br_2$ by consuming other radicals (e.g. $HO_2$) is known as the "bromine
explosion" (Wennberg, 1999).  Ground-based instruments have observed sharp increases in reactive
bromine levels over the course of a single day from below 2 pmol/mol up to a maximum of 41 pmol/mol
(Pöhler et al., 2010). Reactions may also sequester reactive bromine into more stable bromine reservoir
species. BrO may react with nitrogen dioxide ($NO_2$) in R8 to form bromine nitrate ($BrNO_3$), which can
also undergo hydrolysis on a saline surface to form HOBr as in HR3.

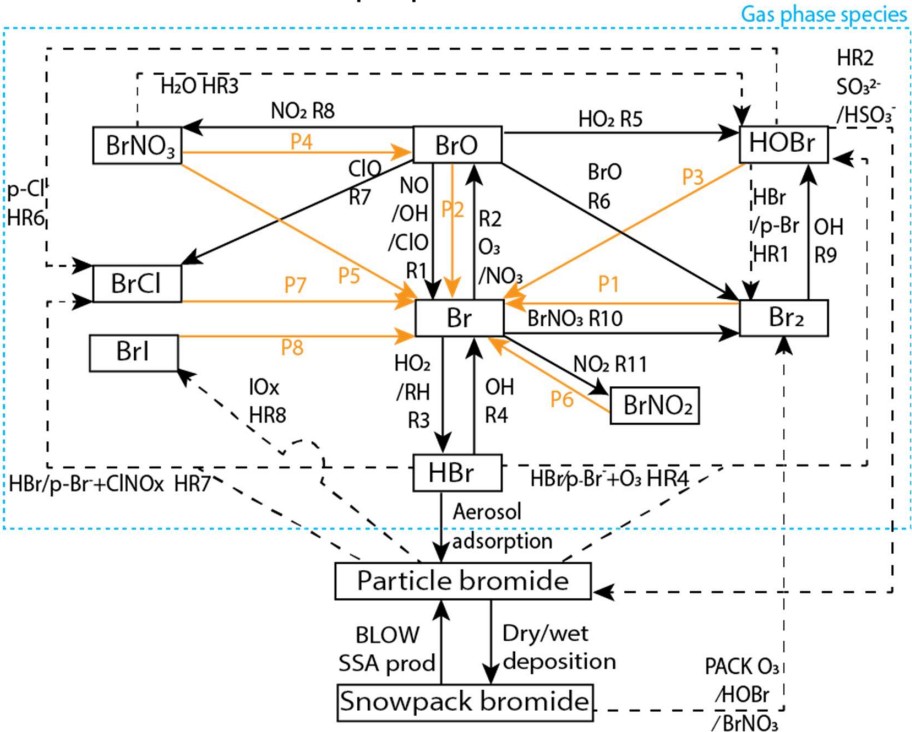


**Figure 1: GEOS-Chem tropospheric bromine reactions.** Tropospheric bromide reservoirs shown in
black boxes, with attached lines indicating reactions. Solid black lines R1-R11 indicate gas phase



chemical reactions, solid orange lines P1-P8 indicate photolysis reactions, and dashed black lines HR1-
HR8 indicate heterogeneous reactions. All gaseous species may undergo wet and dry deposition.
Additional sources of tropospheric bromine include the production of particulate bromide by the BLOW
mechanisms and the production of $Br_2$ by the PACK mechanism, as well as the degradation of
organobromines to form Br (OR1).
A potentially important competitor for recycling of reactive bromine through HOBr is its reaction
with sulfur (IV) species, such as the reaction between $HSO_3^-$ and HOBr in HR2 (Chen et al., 2017).  To
the extent that this reaction competes with HR1, it can slow the release of bromide from surfaces and
reduce gas-phase reactive bromine (e.g., reduce BrO). Deposition of the HBr formed from HOBr by HR2
can remove reactive bromine from the troposphere. In general, the termination of this chemistry leads to
formation of HBr, which undergoes gas-particulate uptake to particulate bromide (p-Br⁻).
Ozone deposited to a saline surface can oxidize Br⁻ to form HOBr (similar to p-Br⁻ reactions
HR4a and HR4b) which is then converted to $Br_2$ or another dihalogen (e.g., BrCl). Production of reactive
bromine during ozone deposition does not require light and can occur at night (Oum et al., 1998; Artiglia
et al., 2017). The production of $Br_2$ is increased at low pH levels (Halfacre et al., 2019).
We define the inorganic bromine family, $Br_y$, in this manuscript as the sum of the bromine
species: Br, BrO, HOBr, $BrNO_3$, $2xBr_2$, BrCl, BrI, and HBr, excluding p-Br⁻. The release of bromine from
sea salt aerosol particles was found to be the dominant global source of reactive bromine (Sander et al.,
2003; Zhu et al., 2019). Sea salt aerosol particles (SSA) sourced from the bursting of bubbles in oceanic
whitecaps and other sources and are one of the most abundant aerosol particle types present in the
troposphere (De Leeuw et al., 2011). Due to their abundance, SSA particles greatly increase the surface
area available for heterogeneous reactive bromine chemistry. Debromination of acidified aerosol
increases reactive bromine by 30%, although global models may underestimate Arctic reactive bromine
when considering only open ocean-sourced SSA (Schmidt et al., 2016). Initial literature on Arctic reactive
bromine chemistry identified aerosol particles as a potential saline surface for reactive bromine
photochemistry (Fan and Jacob, 1992; Vogt et al., 1996). If one supposes that SSA can only be produced
from the open ocean source of SSA, the lack of Arctic Ocean open water during the winter/spring  is at
odds with observations of high SSA concentrations observed during the winter months in polar regions
(Wagenbach et al., 1998; Huang et al., 2018). The formation of SSA from the sublimation of blowing
snow particles over the Arctic Ocean was proposed as an alternate SSA production mechanism (Yang et
al., 2008, 2010, 2019). Recent field studies have confirmed the direct production of SSA from blowing
snow (Frey et al., 2020). A blowing snow mechanism was implemented in the global chemical model
GEOS-Chem and was able to explain wintertime SSA enhancements over the Arctic (Huang and Jaeglé,



2017) as well CALIOP-detected aerosol particle abundance (Huang et al., 2018) and high levels of Arctic
BrO detected by satellites in spring (Huang et al., 2020).
Snowpack containing bromide salts was also identified as a source of reactive bromine (Tang and
McConnell, 1996). Molecular bromine was measured above the snowpack at levels up to 25 pmol/mol
(Foster et al., 2001). Field experiments demonstrate that the snowpack emits $Br_2$, $Cl_2$, and BrCl, with
emission affected by ambient ozone levels, the snowpack ratio of bromide to chloride, and exposure to
sunlight (Pratt et al., 2013; Custard et al., 2017). Box modeling found that the flux of reactive bromine
from the surface of the Arctic Ocean sea ice is a prerequisite for bromine activation (Lehrer et al., 2004).
Box modeling found that both HOBr and $BrNO_3$ can be converted to $Br_2$ in the snowpack (Wang and
Pratt, 2017). Detailed one dimensional models of the snowpack-air interface find that reactive bromine
production can occur in the interstitial air between snowpack grains (Thomas et al., 2011; Toyota et al.,
2014), with ozone depletion events arising from snowpack reactive bromine production (Thomas et al.,
2011; Cao et al., 2016). However, a detailed snowpack model coupled to an atmospheric model would be
sensitive to important parameters such as snowpack bromide content and acidity of the air-ice interface
that are highly variable across the Arctic (Toom-Sauntry and Barrie, 2002; Krnavek et al., 2012). A
mechanism to parameterize the release of molecular bromine from snowpack upon deposition of ozone,
HOBr, and $BrNO_3$ was implemented in the GEM-AQ model and captured many of the observed features
of reactive bromine in the Arctic troposphere (Toyota et al., 2011). The mechanisms from Toyota et al.
(2011) assumes a 100% yield of molecular bromine on deposition of HOBr or $BrNO_3$ (see Figure 1
PACK) and a diurnally varying yield  of $Br_2$ on ozone deposition of 7.5% during the daytime (solar
elevation angle > 5°) and 0.1% during the nighttime (solar elevation angle < 5°) (see Figure 1 PACK). In
the Toyota et al. (2011) parameterization, the daytime yield of $Br_2$ from ozone was increased to 7.5% to
match surface ozone depletion observations and is based on the assumption that photochemical reactions
in the snowpack would trigger a bromine explosion and amplify the net release of $Br_2$ (Toyota et al.,
2011). Herrmann et al (2021) implemented the Toyota et al. (2011) mechanism in WRF-Chem and found
snowpack $Br_2$ production was capable of replicating ozone depletion events observed in multiple datasets.
Marelle et al. (2021) implemented a surface snowpack mechanism based on Toyota et al. (2011) and a
blowing snow mechanism based on Yang et al. (2008) and Huang and Jaeglé (2017) and found improved
prediction of ozone depletion events, the majority of which were triggered by the snowpack mechanism.
The Toyota et al. (2011) mechanism was also implemented in the EMAC model and replicated many of
the features of reactive bromine events observed by satellite-based GOME sensor (Falk and Sinnhuber,

147    2018).



Field campaigns have directly observed the production of SSA from blowing snow (Frey et al.,
2020) as well as production of $Br_2$ from the snowpack (Pratt et al., 2013) in the environment. This
manuscript uses both production mechanisms for the first time in the global chemical model GEOS-
Chem. We devised a set of six model runs to test each mechanism individually and together as well as one
control run using neither mechanism. We compare BrO simulated in each model run against extensive
ground-based observations of BrO made from February to June 2015. This set of modeling scenarios
allows identification of the effects of each mechanism on BrO as well as the synergistic effects of both
mechanisms working together.

## 156  2.    Data sources and methods

### 157  2.1 MAX-DOAS observation platforms

Multiple axis differential optical absorption spectroscopy (MAX-DOAS) remotely measures the
vertical profile of BrO (Hönninger and Platt, 2002; Carlson et al., 2010; Frieβ et al., 2011; Peterson et al.,
2015; Simpson et al., 2017). BrO is commonly used as a proxy for total tropospheric reactive bromine
(Chance, 1998; Richter et al., 1998; Wagner and Platt, 1998; Theys et al., 2011; Choi et al., 2012). MAX-
DOAS instruments were mounted on all of the fifteen floating autonomous platforms (O-Buoys) deployed
in the Arctic sea ice as a part of the National Science Foundation-funded Arctic Observing Network
project (Knepp et al., 2010). Since MAX-DOAS requires sunlight to operate, measurements are not
available in winter. Spring observations on the O-Buoys typically begin in April when there is enough O-
Buoy solar power to defrost the MAX-DOAS viewport. Figure 2 shows the O-Buoys active during 2015.
O-Buoy 10 was deployed into sea ice in fall 2013 and measured reactive halogen chemistry in spring
2014 and 2015.  Most O-Buoys were destroyed in the summer, crushed between fragments of melting sea
ice.  However, O-Buoy 10 survived summer 2014 in an intact ice floe, survived the winter of 2014-15,
and re-started MAX-DOAS observations in April 2015. O-Buoys 11 and 12 were deployed in fall 2014
and also re-started observing BrO in April 2015. Figure 2 shows the GPS-derived tracks of the O-Buoys
for their full deployment and highlights the O-Buoy locations from April to June 2015 when the BrO
observations considered in this analysis were gathered. A MAX-DOAS instrument of the same design
was deployed at the Barrow Arctic Research Center (BARC) on the coast of the Arctic Ocean located at
156.6679°W, 71.3249°N near Utqiaġvik, AK (Simpson, 2018), also shown in Figure 2.  Unlike the O-
Buoy MAX-DOAS systems, which were powered by batteries and solar panels, the BARC MAX-DOAS
was powered from local utilities and was able to defrost its viewport to gather BrO observations earlier in
the year, including February and March 2015. The BARC MAX-DOAS data was compared with two O-
Buoy style MAX-DOAS instruments deployed on Icelander platforms (deployed on top of sea ice instead
of within) and measurements from the various MAX-DOAS systems were found to be comparable
(Simpson et al., 2017). The reactive bromine season ends when the BrO slant column densities fall below





the instrument detection limit and do not recover, which we call the seasonal end date (Burd et al., 2017).
All O-Buoy and BARC (Utqiaġvik) data are available at arcticdata.io (Simpson et al., 2009) (Simpson,
2018). More information on the time periods of spring BrO observations can be found in Swanson et al.
(2020) and Burd et al. (2017). For comparison to the MAX-DOAS BrO observations, GEOS-Chem model
simulations are sampled along the GPS-derived paths of O-Buoys 10, 11 and 12 as well as at BARC.

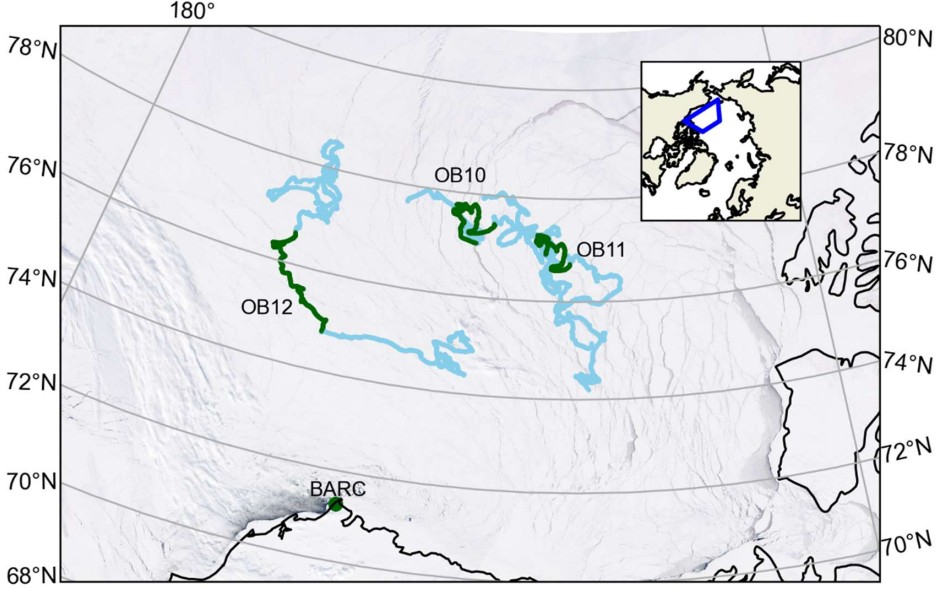


**Figure 2: Locations of MAX-DOAS BrO observations used in this work.**
Blue lines show the drift tracks of O-Buoys, with green showing the locations with valid BrO
measurements in spring 2015. Location of Barrow Arctic Research Center (BARC) in Utqiaġvik indicated
by green dot. Inset map shows True color MODIS imagery on 1 April 2015 shows typical sea ice
coverage (NASA 2015).



## 2.2 MAX-DOAS profile retrieval


Vertical profiles of BrO were derived from MAX-DOAS observations by means of optimal

estimation inversion procedures detailed in Peterson et al. (2015) with settings detailed in Simpson et al.
(2017). The HeiPro optimal estimation algorithm (Frieß et al., 2006, 2019) is used to retrieve a vertical
profiles of BrO between the surface and 4km from the MAX-DOAS observations. Examination of the
averaging kernels from each MAX-DOAS retrieval finds the retrieved vertical profile of BrO is best
represented by two quantities: the vertical column density of BrO in the lowest 200 m, and the vertical
column density of BrO in the lowest 2000 m of the troposphere referred to in this manuscript as $BrO_{LTcol}$
(Peterson et al., 2015). We approximate surface mixing ratio by assuming well mixed constant
distribution of BrO throughout the lowest 200 m. This mixing ratio is reported as $BrO_{pptv200}$ (Simpson et al
2009, Simpson 2018). It was shown in Peterson et al. (2015) that these two quantities were largely
independent of each other, were fairly insensitive to variations in the assumed prior profile, and
represented the ~2-3 degrees of freedom for signal indicated by the optimal estimation retrieval. An
important consideration of this method is that when the visibility is poor, the MAX-DOAS is unable to
traverse the lowest 2000m AGL and the $BrO_{LTcol}$ cannot be measured accurately. Therefore, our quality-
control algorithm eliminates $BrO_{LTcol}$ observations when the degrees of freedom for signal in the lofted
(200m - 2000m AGL) layer were below 0.5 (Simpson et al., 2017). The average fitting error (1σ error) of
$BrO_{LTcol}$ during spring 2015 was $5.6*10^{12}$ molecules/cm$^2$.

## 2.3 SSA production from open ocean


Seafoam from breaking waves and bursting of bubbles forms aerosol droplets suspended in the

marine boundary layer (Lewis and Schwartz, 2004). We calculate emission of sea salt aerosol particles
from the open ocean as a function of wind speed and sea surface temperature (SST) using the mechanism
initially described in Jaeglé et al. (2011) and updated with decreased emissions over cold (SST < 5°C)
ocean waters (Huang and Jaeglé, 2017). Two separate SSA tracers are transported: accumulation mode
SSA ($r_{dry}$ = 0.01−0.5 μm) and coarse mode SSA ($r_{dry}$ = 0.5–8 μm). Sea salt bromide is emitted assuming
bromine content of $2.11×10^{-3}$ kg Br per kg of dry SSA (primarily NaCl) based on the mean ionic
composition of sea water (Sander et al., 2003). Bromide content is tracked separately on accumulation
mode SSA and on coarse mode SSA. Heterogeneous chemical reactions can convert SSA-transported
bromide into gaseous reactive bromine species in the atmosphere. We run our open ocean SSA
calculations at 0.5° latitude x 0.625° longitude spatial resolution using the harmonized emissions
component (HEMCO) for highest possible detail (Keller et al., 2014; Lin et al., 2021) including cold
water corrections used in Jaeglé et al. (2011) . Production of SSA from open oceans which can lead to
Arctic reactive bromine recycling on advected open ocean SSA within GEOS-Chem. Each of our model



runs reads the dataset generated offline by HEMCO rather than spend computational time replicating
open ocean SSA emissions. We call our control run using only open ocean SSA emissions BASE.

**228    2.4 Blowing snow SSA production**

Snow can be lofted from the snowpack into the lowest layers of the troposphere by high

windspeeds, where it can undergo saltation (bouncing leading to fragmentation) and sublimation to form
SSA (Yang et al., 2008, 2010; Frey et al., 2020). This process is modeled as a function of humidity,
ambient temperature, windspeed, and the salinity of the blowing snow (Yang et al., 2008, 2010). Three
thresholds must be met for SSA production from blowing snow (Dery and Yau, 1999; Déry and Yau,
2001). A temperature threshold restricts SSA production from blowing snow to temperatures below
freezing. The humidity threshold is based on relative humidity with respect to ice. Sublimation from snow
crystals cannot occur if the air is saturated, and no SSA is produced if $RH_{ice}$ is greater than 100%. The
windspeed threshold requires ten-meter wind speed to be greater than a threshold value defined in
Equation 1 for any production of SSA (Dery and Yau, 1999; Déry and Yau, 2001).
$U_t \ = \ 6.975 \ + \ 0.0033(T_s \ + \ 27.27)^2$                                       (1)
The wind speed threshold ($U_t$) is dependent on surface temperature ($T_s$) in Celsius with a minimum
threshold of 6.975 m/s at -27.27 C° and a maximum threshold at 0 C° of 9.429 m/s. The ten-meter
windspeed threshold is the most stringent and often controls the production of SSA from blowing snow.

Production of blowing snow and SSA is highly sensitive to surface windspeed. We use the

highest resolution surface windspeed dataset to ensure the most accurate modeling of SSA and reactive
bromine. The MERRA-2 Global Reanalysis Product has a 0.5° latitude x 0.625° longitude resolution
which is typically re-gridded to a lower resolution for global chemical modeling. Previous use of the
snowpack blowing snow mechanism has simulated blowing snow with MERRA-2 data re-gridded to
either 2°x2.5° or 4°x5° latitude and longitude (Huang and Jaeglé, 2017; Huang et al., 2018, 2020). Re-
gridding to coarser spatial resolution may smooth out the highest ten-meter windspeeds by averaging
them with lower windspeeds in the grid cell. The Utqiaġvik MERRA-2 ten-meter windspeeds at different
spatial resolutions are shown in Supplemental Figures S1, S2 and S3 to illustrate this effect.  Average
Utqiaġvik ten-meter windspeeds for 2015 are 5.3 m/s at 2°x2.5° resolution and 5.5 m/s at 0.5°x0.625°
resolution. The maximum Utqiaġvik ten meter windspeed at MERRA-2 2x2.5 is 16.3 m/s, while the
maximum windspeed at MERRA-2 0.5°x0.625° is 19.3 m/s. These extremely high windspeed events are
more common at higher spatial resolution and can contribute an outsized amount of SSA to the marine
boundary layer. Supplemental Figure S4 shows the measured ten-meter windspeed at BARC, along with
daily average threshold windspeed (Equation 1). Spikes in daily averaged windspeed at BARC in April
can contribute to SSA formation and justify the use of high-resolution MERRA-2 wind speed data.





Snow salinity is influenced by snow age and the material underlying the snow (Krnavek et al.,
2012). The median surface snowpack salinity near Utqiaġvik was measured at 0.67 practical salinity units
(PSU)PSU for 2-3 weeks old sea ice, 0.12 PSU for thicker first year ice, and 0.01 PSU for multi-year ice
(MYI) (Krnavek et al., 2012). Snow salinity is also a function of snow depth above sea ice, with blowing
surface snow having much lower salinity than snow at depth that is in contact with the sea ice (Frey et al.,
2020). Domine et al. (2004) measured median salinity at 0.1 PSU on snowpack over first year ice and
0.02 PSU on snowpack over multi-year ice. In this analysis we use a salinity of 0.1 PSU on first-year sea
ice as in Huang et al. (2020). The production of reactive bromine from sea ice types is entirely dependent
on PSU in this parameterization. Previous modeling efforts have used 0.01 PSU for MYI (Huang et al.,
2018) and underestimate BrO production in high Arctic areas with increased MYI coverage. The bromide
content of surface snow over MYI is enriched by deposition of SSA and trace gases, and MYI regions
may play a role in springtime halogen chemistry (Peterson et al., 2019). Previous analysis of O-Buoy data
found no statistically significant differences in springtime BrO between regions of the Arctic (Swanson et
al., 2020). We use 0.05 PSU for snowpack on MYI as in Huang et al. (2020).
Another important parameter for SSA formation is the number of SSA particles formed from each
blowing snowflake. A value of 5 particles per snowflake was used in Huang and Jaeglé (2017) based on
wintertime observations of supermicron and sub-micron SSA at Barrow. Values of 1 and 20 particles per
snowflake have been tested (Yang et al., 2019) but it is unclear which value was more realistic. We use a
particle formation value of 5 particles per snow grain as in Huang et al. (2020).
Snowpack may be enriched or depleted in bromide compared to seawater, which is thought to be
an effect of atmospheric deposition or release of bromine from snowpack (Krnavek et al., 2012).
Snowpack enrichment due to atmospheric deposition is less pronounced when snowpack salinity is high,
with snowpack containing 1000 μM $Na^+$ (approximately 0.06 PSU) or more never exceeding twice the
seawater ratio of bromine to chloride (Krnavek et al., 2012). Domine et al. (2004) found an increased
enrichment factor of five times seawater in snow with a salinity of 100 μM $Cl^-$ (approximately 0.006
PSU). We use a snowpack enrichment factor of bromide five times that of seawater as in Huang et al.
(2020) where this enrichment best agreed with GOME-2 observations. However, we note that a bromide
enrichment factors five times seawater exceeds enrichment factors of two measured in snowpack with a
salinity of 0.1 PSU (Krnavek et al., 2012).
Our choice of model input settings is similar to Huang et al. (2020) but we will be running the
blowing snow mechanism in HEMCO at a 0.5° latitude x 0.625° longitude spatial resolution. The model
run using the results of our high-resolution blowing snow SSA HEMCO simulation is called BLOW.



### 2.5 Snowpack emissions of molecular bromine


We base our $Br_2$ emissions scheme on Toyota et al. (2011) and Marelle et al. (2021), which

prescribe a yield of $Br_2$ upon snowpack deposition of ozone, $BrNO_3$ and $HOBr$. In other modeling studies,
this simplified deposition-based mechanism captured the synoptic-scale behavior of reactive bromine
production across the Arctic (Toyota et al., 2011; Falk and Sinnhuber, 2018; Herrmann et al., 2021;
Marelle et al., 2021). These modeling studies used different yields of $Br_2$ upon deposition over land
snowpack, multi-year ice, and first year ice, restricting the production of molecular bromine from ozone
deposition to first year ice surfaces. None of these studies were coupled to a snowpack model tracking
snow bromide, and effectively assume an infinite bromide reservoir with $Br_2$ production limited only by
the deposition flux and $Br_2$ yield.

Field studies indicate that snowpack over multi-year ice, first-year ice, and land regions may

contribute to reactive bromine chemistry. Krnavek et al. (2012) found snow bromide content spanning six
orders of magnitude, with individual samples taken from multi-year ice, first-year ice, and land regions
showing variability of up to three orders of magnitude for each region. Analysis of variance in
tropospheric BrO from 2011-2016 found no statistically significant differences in tropospheric BrO
between different regions of the Arctic (Swanson et al., 2020). Both coastal snowpack and multi-year ice
regions may produce reactive bromine. Molecular bromine production has been observed from coastal
snowpack on exposure to ozone (Pratt et al., 2013; Custard et al., 2017). Airborne sampling has observed
enhanced BrO up to 200 km inland (Peterson et al., 2018). Snow above multi-year sea ice regions is
depleted in bromide, indicating that it may play a role in Arctic bromine chemistry (Peterson et al., 2019).

Our modeling study tests the hypothesis that all snow has a uniform ability to produce molecular

bromine, effectively assuming an infinite bromide reservoir with $Br_2$ production limited only by the
deposition flux. We differ from previous model parameterizations in allowing uniform $Br_2$ production
upon snowpack deposition of ozone, $BrNO_3$ and $HOBr$ over all sea ice surfaces and selected coastal
snowpack regions. We expect higher predictions of snowpack molecular bromine production than recent
modeling efforts (Herrmann et al., 2021; Marelle et al., 2021) in which ozone deposition over land and
multi-year ice surfaces did not produce molecular bromine.

### 2.5.1 Snowpack $Br_2$ production over sea ice


We assume a uniform production of $Br_2$ on deposition to snowpack over oceanic ice whether the

ice is first-year sea or multi-year sea ice. We use MERRA-2 fractional ocean ice coverage fields, which
introduces some artifacts. MERRA-2 classifies the freshwater Great Lakes as ocean, but sea ice and
snowpack on those frozen lakes is unlikely to have sufficient bromide to support large $Br_2$ fluxes due to
its distance from the ocean. Therefore, we specifically prohibit snowpack $Br_2$ emissions in the Great



Lakes region (between 41° N and 49° N latitude and 75° W and 93° W longitude). This choice is in
agreement with McNamara et al. (2020), who found road salt derived aerosol particles are responsible for
80-100% of atmospheric $ClNO_2$ in Michigan with no mention of a source of reactive halogens from
nearby Great Lakes.

**2.5.2 Snowpack $Br_2$ production over land**

We wish to only enable production of $Br_2$ over land if the snowpack is sufficiently enriched in

bromide. Snowpack over land surfaces and glaciers may be enriched in bromide by oceanic SSA sources
(Jacobi et al., 2012, 2019). The distance that SSA may be transported inland from the coast is limited by
geographical features such as mountains. Based on direct observations of reactive bromine chemistry up
to 200 km from the Alaskan coastline (Peterson et al., 2018), we include unlimited production of $Br_2$ from
specific land grid cells within 200 km of the coast upon deposition of ozone, HOBr, and $BrNO_3$. We only
allow the fraction of each grid cell that is within 200 km of the coastline (Group and Stumpf, 2021) to
produce molecular bromine. We further restrict snowpack $Br_2$ emissions to locations that are less than 500
m above sea level, because higher elevation locations are unlikely to be enriched by sea spray. This
altitude screen eliminates $Br_2$ emissions from coastal mountains such as the Alaskan Rockies, the Brooks
Range in Alaska, and the Scandinavian Mountains as well as from the Greenland Plateau. Halogen
chemistry may occur over the Greenland ice sheet (Stutz et al., 2011) contrary to this screen, but this will
have minimal impact on the regions of interest in this manuscript.

Our final screen is based on the average snow depth in each land grid cell. Both modeling studies

(Thomas et al., 2011; Toyota et al., 2014) and field studies (Domine et al., 2004; Pratt et al., 2013;
Custard et al., 2017; Frey et al., 2020) agree that bromine chemistry can occur in the better ventilated and
illuminated top  of the snowpack. Regions with less than 10 cm of snowpack may not have sufficient
snow for reactive bromine chemistry, thus we only produce snowpack $Br_2$ when the average snow depth
in a land grid cell is 10 cm or greater. This screen prevents molecular bromine production in the lower
latitude regions with minimal snow coverage and is necessary because ozone deposition to plants in
snow-free grid cells often exceeds the slow deposition of ozone to snowpack and would not be expected
to produce $Br_2$.

**2.5.3 Diurnal yield of $Br_2$ on ozone deposition**

We choose two alternate assumptions for the yield of $Br_2$ during the day. Toyota et al. (2011)

initially assumed a constant yield of $Br_2$ from ozone deposition of 0.1% based on laboratory observations
of nighttime bromine activation on ozone deposition (Oum et al., 1998; Wren et al., 2010, 2013) and then
adjusted the daytime yield of $Br_2$ on ozone deposition to 7.5% to better match surface ozone mixing ratios
measured at coastal stations. This increased daytime yield value was chosen based on the assumption that





photochemistry may trigger an autocatalytic cycle leading to a 75-fold increase in $Br_2$ yield. The
PHOTOPACK runs uses the increased daytime $Br_2$ yield of 7.5% when the solar elevation angle is 5° or
greater. Previous implementations of the snowpack mechanism (Toyota et al., 2011; Herrmann et al.,
2021; Marelle et al., 2021) predict ozone deposition velocities over Arctic sea ice on the order of 0.01
cm/s. Our model predicts similar ozone deposition rates over polar open ocean of 0.009 cm/s (Pound et
al., 2020), but our model currently predicts deposition velocities over Arctic sea ice between 0.02 cm/s
and 0.1 cm/s based on the month (see Supplemental Figure S5), with higher values influenced by
proximity to the coast as described in Bariteau et al. (2010).. Thus, our PHOTOPACK run may predict
much higher Br emissions than previous snowpack predictions despite the same yield values due to
differences in deposition. To match out magnitude of $Br_2$ production with previous implementations of
the snowpack mechanism (Toyota et al., 2011; Herrmann et al., 2021; Marelle et al., 2021) we add two
PACK runs with a constant $Br_2$ yield on ozone deposition of 0.1% based on yield values in Toyota et al.
(2011). Both PACK and PHOTOPACK runs assume 100% conversion of deposited HOBr and $BrNO_3$ to
$Br_2$. Table 1 shows further model run yield details.
**Table 1 Model run settings**
Sea salt aerosol particles are produced from blowing snow as detailed in Section 2.5. Daytime is defined
as when the solar elevation angle is greater than 5°, nighttime is defined as when the solar elevation angle
is less than 5°.

| Model Run | Blowing snow SSA produced | Millimoles Br yielded per mole O3 deposited (daytime) | Millimoles Br yielded per mole O3 deposited (nighttime) |
|---|---|---|---|
| BASE | FALSE | 0 | 0 |
| BLOW | TRUE | 0 | 0 |
| PACK | FALSE | 1 | 1 |
| BLOW+PACK | TRUE | 1 | 1 |
| PHOTOPACK | FALSE | 75 | 1 |
| BLOW+PHOTOPACK | TRUE | 75 | 1 |


**2.6 GEOS-Chem chemistry and transport model**
The GEOS-Chem global atmospheric chemistry and transport model (Bey et al., 2001) simulates
emissions, transport, and chemistry of atmospheric trace gases and aerosols, including halogens. The
chemical mechanism in GEOS-Chem 12.9.3 (http://www.geos-chem.org, last access 29 October 2019,
DOI:10.5281/zenodo.3974569) includes $HO_x$-$NO_x$-VOC-$O_3$-halogen-aerosol tropospheric chemistry
(Mao et al., 2013; Fischer et al., 2014; Fisher et al., 2016; Travis et al., 2016; Wang et al., 2021). The
model has been regularly and consistently updated to reflect current understanding of heterogeneous and
gas-phase halogen chemistry.



Halogens in the troposphere may be sourced from photooxidation of halocarbons, emissions of
iodine from the ocean surface, downward transport of halogens from the stratosphere, and release of
halogens through heterogeneous chemistry on SSA.  Figure 1 shows a simplified version of the GEOS-
Chem reaction scheme focusing on tropospheric bromine reactions and reservoirs. Heterogeneous
reactions for release of reactive bromine from aerosol surfaces were added to GEOS-Chem (Parrella et
al., 2012) and have been updated to include multiphase reactions and reactions between bromine, chlorine
and iodine species (Schmidt et al., 2016; Sherwen et al., 2016a; Wang et al., 2019b) as well as input from
the stratosphere (Eastham et al., 2014). Recent updates also include reactions between sulfur (IV) species
and HOBr, which lead to a 50% decrease in $Br_y$ due to the scavenging of HOBr on aerosol surfaces
containing sulfur (Chen et al., 2017). These HOBr-sulfur(IV) reactions are critical in moderating
tropospheric BrO in the mid latitudes (Zhu et al., 2019). In GEOS-Chem 12.9 the halogen chemical
mechanism was modified extensively to include chlorine chemistry as detailed in Wang et al. (2019b)
with update halogen-sulfur (IV) rates (Liu et al., 2021) as well as improved cloud pH calculation from
Shah et al. (2020).  For the simulations here, GEOS-Chem uses the Modern-Era Retrospective Analysis
for Research and Applications, version 2 (MERRA-2) assimilated meteorological fields (Gelaro et al.,
2017) re-gridded from native resolution of 0.5°x0.625° latitude and longitude to 2°x2.5° using a reduced
vertical grid of 47 layers.
We initialize our model in October 2014 from a full-chemistry benchmark file, allowing for 6
months of spinup before our period of interest spanning from March to November 2015. We run six
different model simulations with settings detailed in Table 1. The base model (BASE) includes the
halogen sources described above but no Arctic-specific halogen sources. The BLOW simulation adds
SSA production from blowing snow following Huang et al (2020) but using a more recent version of
GEOS-Chem. The PACK simulation adds snowpack $Br_2$ emissions using a constant yield from $O_3$
deposition. The PHOTOPACK simulation also emits $Br_2$ from snowpack but increases the $Br_2$ yield from
$O_3$ deposition under sunlight. These blowing snow and snowpack sources are combined in the
BLOW+PACK and BLOW+PHOTOPACK simulations.

**2.7 Comparing GEOS-Chem results to MAX-DOAS vertical column densities**


GEOS-Chem simulates BrO mixing ratios for each of its 47 atmospheric layers.  Reducing the
vertical resolution of the more-resolved GEOS-Chem predictions to be comparable to the coarser MAX-
DOAS data is necessary for appropriate comparison (Rodgers and Connor, 2003).  To compare the
GEOS-Chem profiles with these two grid-coarsened quantities, we grid-coarsen the averaging kernels
produced by the HeiPro retrieval algorithm using Supplemental Equation S1 from Payne et al. (2009) to
the partial column averaging kernels shown in Figure 3. We use the average of all April averaging kernels





that pass our quality criteria (>0.5 DOFS in the lofted layer), which generally represents non-cloudy
conditions.  We calculate modeled $BrO_{LTcol}$ by applying the partial column averaging kernels shown in
Figure 3 to the GEOS-Chem modeled vertical BrO profiles.
Figure 3 shows the average partial column averaging kernel for the surface layer (0-200m AGL)
has near unit sensitivity to BrO at the ground, decaying to about 0.5 at 200m AGL then to zero at about
400m AGL, as desired.  The sensitivity of the $BrO_{LTcol}$ is near unity from about the surface to 600m AGL,
then slowly decays with 0.5 sensitivity at 2000m AGL. The resulting sensitivity to mid-tropospheric BrO
means that free-tropospheric BrO produced by the GEOS-Chem model contributes to modeled $BrO_{LTcol}$,
albeit at 50% or lower sensitivity, even if the GEOS-Chem-predicted free-tropospheric BrO is above the
nominal 2000m top of the integration window.  The residual sensitivity of the $BrO_{LTcol}$ averaging kernel
above 2000m is caused by the limited ability of ground-based MAX-DOAS to distinguish the true altitude
of BrO at non-tangent geometries (higher viewing elevation angles) that are required to view BrO at these
higher altitudes. Figure 3 shows that BrO above 4 km makes only a small contribution to the modeled
$BrO_{LTcol}$, which was not included in the $BrO_{LTcol}$.

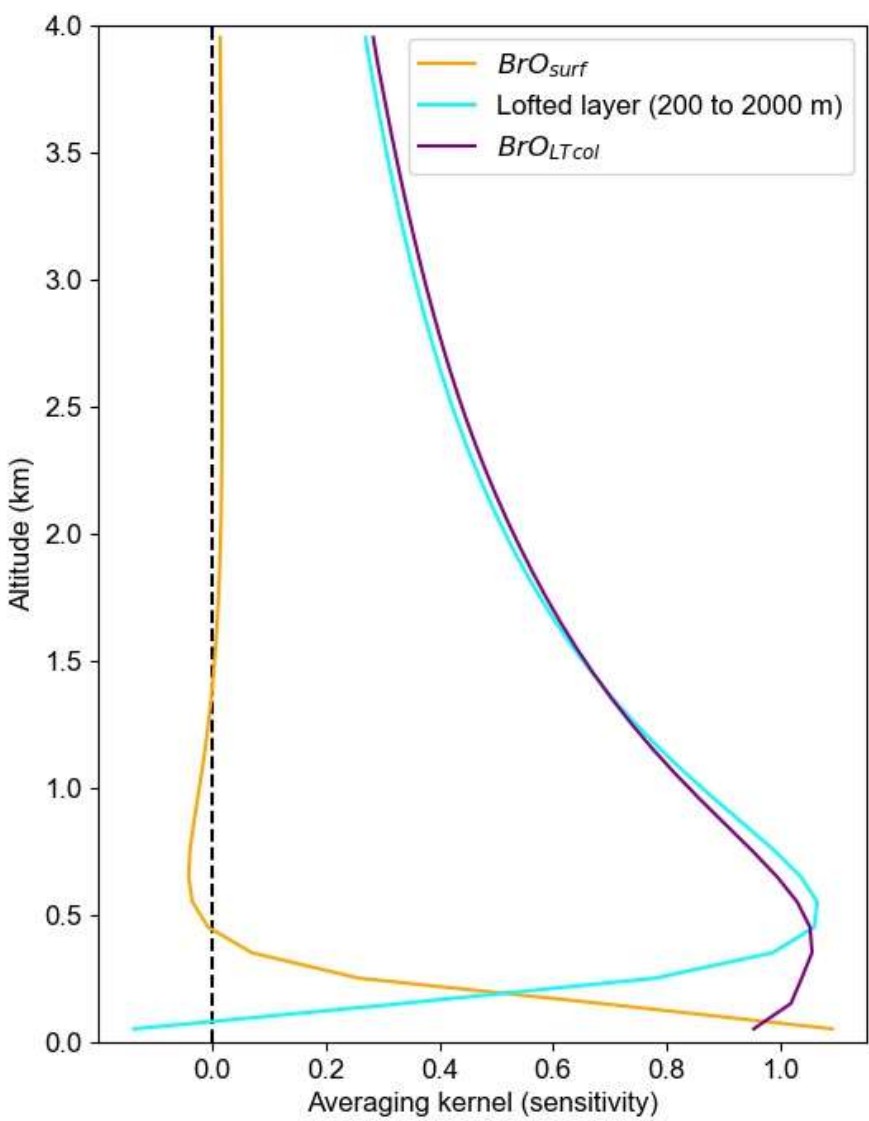


**Figure 3: Averaging kernels showing the sensitivity of retrieved BrO$_{LTcol}$ and retrieved BrO$_{surf}$ to BrO at a range of altitudes.**

Each line represents a row of the averaging kernel matrix. BrO$_{surf}$ is the column from the surface to 200 m and BrO$_{LTcol}$ is the column up to 2000 m.

Although it has been suggested in the literature (von Clarmann and Glatthor, 2019) that averaged averaging kernels can cause problems, we do not report data when there are clouds and thus are only using the more consistent averaging kernels that occur under clear sky conditions. We use other criteria related to vertical visibility to identify clear skies. As described in Peterson et al. (2015), the information





content (DOFS) in the lofted layer is nearly linearly related to the aerosol optical depth.   We find that the
slant column density of the $O_2$-$O_2$ collisional dimer (aka $O_4$) observed at 20° elevation angle is correlated
with the lofted DOFS (Supplemental Figure S6).  From this correlation we find that clear sky conditions
have 20° elevation angle $O_4$ dSCD > $10^{43}$ molecule$^2$cm$^{-5}$ and use this cut to distinguish clear sky versus
clouds.  To assure that GEOS-Chem results are only compared to the clear-sky observational data, we
apply this clear sky screen to the modeled BrO$_{LTcol}$ timeseries.  The use of this screen also assists in
minimizing variability in the averaging kernels and thus allowing the April averaged partial column
averaging kernels (Figure 3) to be applied for clear skies at any time of the year.

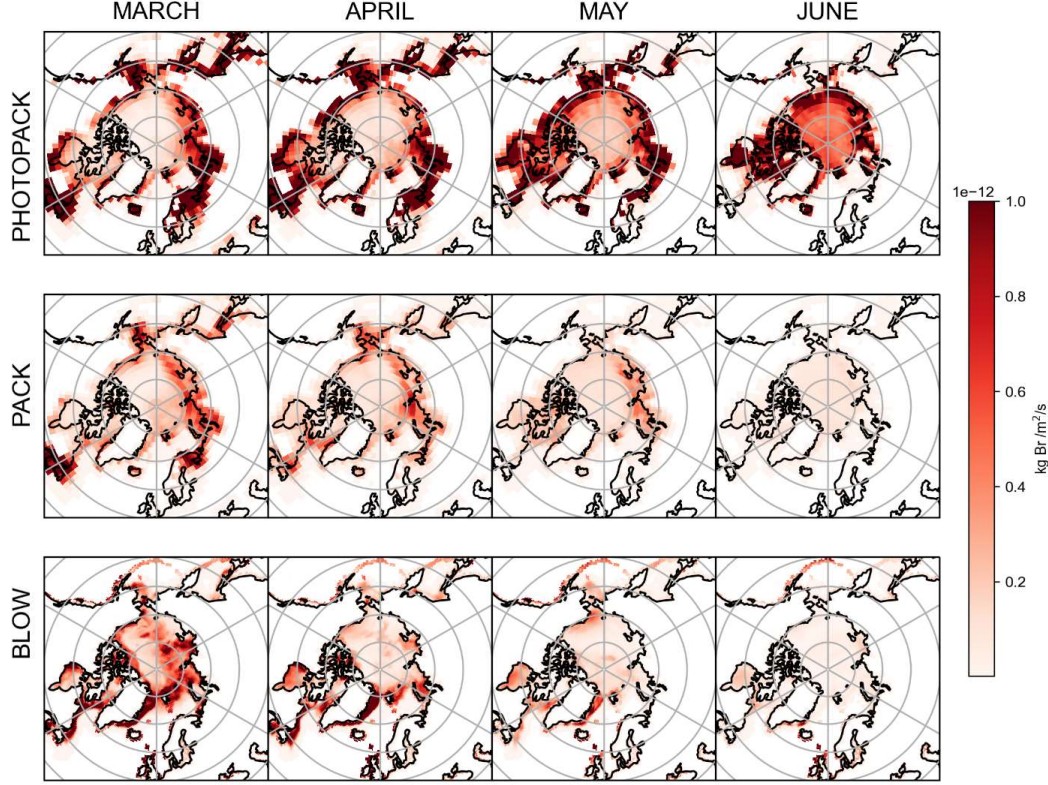


**Figure 4: Mean snowpack Br$_2$ emissions and p-Br$^-$ by month, as simulated by GEOS-Chem.**
The top row shows emissions of Br$_2$ in the PHOTOPACK run, the middle row shows the emissions of
Br$_2$ in the PACK run, and the bottom row shows emissions of p-Br$^-$ from adding the BLOW mechanism.


## 3. Examining reactive bromine in the Arctic spring

### 3.1 Snowpack Br$_2$ emissions

The top two rows of Figure 4 shows PHOTOPACK and PACK average snowpack Br$_2$ emissions for each spring month. The emission of Br$_2$ in PHOTOPACK increases over the Arctic Ocean in May and June, when the sun is above the horizon for up to 24 hours per day and ozone deposition yield is almost always at the photo-enhanced level of 7.5%. Notably, Br$_2$ emissions over the Arctic Ocean in the PHOTOPACK and BLOW+PHOTOPACK runs are highest in June when the sun is nearly always five degrees above the horizon and surface temperatures may drop below freezing. The PACK emissions are lower than the PHOTOPACK Br$_2$ emissions by an order of magnitude and shows a seasonal cycle with a high BrO$_{LTcol}$ in April and May with a decrease in May and June. While our ozone deposition velocities (see Supplemental Figure S5) over Arctic sea ice are much higher than previous estimates of an approximate magnitude of 0.01 cm/s (Toyota et al., 2011), the PHOTOPACK run highlights that a 75-fold increase in daytime Br$_2$ yield can lead to predictions of increased Br$_2$ production over the North Pole in June. Monthly satellite observations show that BrO reaches a minimum over the Arctic Ocean in June (Richter et al., 1998).

Coastal land regions within 200 km of the coastline have some of the highest modeled snowpack Br$_2$ emissions (see Figure 4 rows 1 and 2). Dry deposition velocities are lower over ice covered ocean than open ocean due to the higher likelihood of a stable surface boundary layer over the ice-covered ocean (Toyota et al., 2016). This remains true within GEOS-Chem, as deposition rates are greatest over land, less rapid over ice-covered ocean, and lowest over open ocean (see Supplemental Figure S5). Lower dry deposition velocities over the ice-covered Arctic Ocean lead to decreased deposition and conversion to Br$_2$. In GEOS-Chem, ozone mixing ratios and deposition are over three orders of magnitude larger than BrNO$_3$ and HOBr mixing ratios and deposition over the Arctic Ocean, and ozone contributes more than half of total Br$_2$ emitted in the PACK and BLOW+PACK runs. Our snowpack mechanism assumes that all ozone deposited to the surface of a grid cell reacts with the snowpack cover and produces Br$_2$. This assumption is more appropriate in the barren snow-covered coastal tundra but may be less accurate in areas where deposition to vegetation dominates. This nonconservative approach may lead to overestimation of Br$_2$ emissions from snowy vegetated surfaces. Our screens for snowpack emissions described in section 1.3.5 tried to minimize these effects but may not work perfectly due to finite grid cell resolution and other challenges. Increased Br$_2$ emissions observed in Figure 4 in northern Europe may also be partially driven by increased local mixing ratios of ozone and NO$_x$ over industrialized regions such as the Kola Peninsula.



### 3.2 Blowing Snow pBr⁻ emissions

The bottom row of Figure 4 shows the total quantity of particulate bromide released by the

blowing snow SSA mechanism in the BLOW runs. Emissions over the Arctic Ocean decline each month
after the March maximum as rising temperatures increase the windspeed threshold for blowing snow SSA
production. Some icy coastal regions with frequently high windspeeds such as the Aleutian Islands south
of Alaska and the eastern coast of Greenland continue to emit SSA p-Br⁻ in April, and the extremely high
winds in the Aleutians enable SSA production into May. The location of specific high-wind storm
systems in spring 2015 may be evident in the darker red spots over the Arctic Ocean, which are
particularly noticeable over the Eurasian and Central Arctic in March. These monthly averages are only
accurate for the months in spring 2015 and may not be spatially representative of blowing snow SSA
production in other years.

The impact of the blowing snow SSA emissions is minimal on O-Buoys in the Beaufort Gyre,

possibly due to the spatial and seasonal variations in SSA p-Br⁻ emissions. Figure 4 shows that 2015 SSA
production was highest in March and April on the Eurasian and Central sector of the Arctic, and thus the
O-Buoys deployed as shown in Figure 2 are less exposed to the effects of SSA production than the Arctic
as a whole. Particulate bromide must be activated from SSA by heterogeneous reactions as in Figure 1,
leading to photochemical cycles that sustain further activation of bromide from SSA. The dearth of
sunlight over the Arctic Ocean in early March coincides with the greatest SSA p-Br⁻ production and
means that the increased February SSA p-Br⁻ emissions may not lead to a direct increase in BrO.

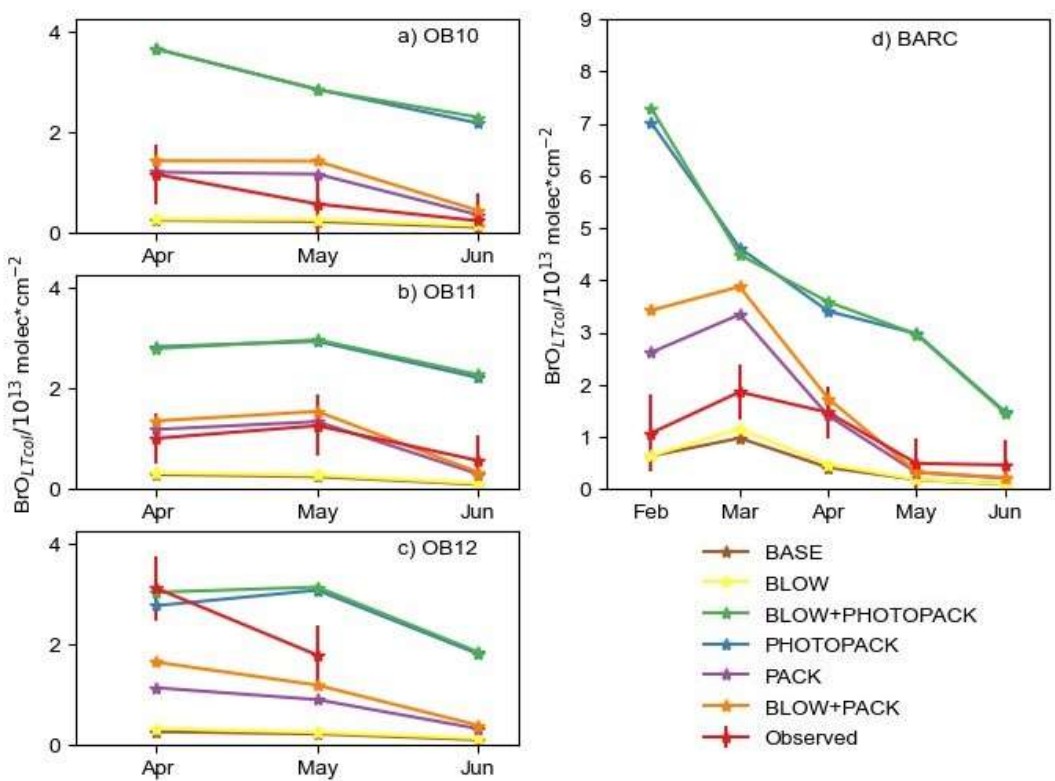

**Figure 5: Monthly average BrO$_{LTcol}$ in observations and model**

Monthly averages of BrO at a) O-Buoy 10, b) O-Buoy 11, c) O-Buoy 12, and d) BARC at Utqiaġvik only using predictions and observations when dSCDO$_4$ >1*10$^{43}$ molecules$^2$cm$^{-5}$. Observations with average 1σ error shown in red. All units in 10$^{13}$ molecules/cm$^2$.

**3.3 Snowpack Br$_2$ emissions have more impact than blowing snow on monthly BrO abundance**

Increased levels of bromine have been historically seen at Utqiaġvik during February, March, April and May (Berg et al., 1983). Previous O-Buoy data analysis noted BrO dropping to zero in June (Burd et al., 2017). Figure 5 shows monthly averaged modeled BrO$_{LTcol}$ at Utqiaġvik and on the O-Buoys for each model configuration. The difference in GEOS-Chem modeled monthly averaged BrO$_{LTcol}$ for O-Buoys is minimal between the BASE and BLOW runs, the PHOTOPACK and BLOW+PHOTOPACK runs, and the PACK and BLOW+PACK runs.

Both BASE and BLOW runs predict near-zero BrO$_{LTcol}$ on all O-Buoys and during most months at Utqiaġvik. The exception to this is the slight increases in monthly modeled BrO$_{LTcol}$ to 1*10$^{13}$ molecules/cm$^2$ in March and April. This BASE increase in BrO$_{LTcol}$ indicates that oceanic SSA rather than blowing snow SSA can affect modeled BrO at Utqiaġvik due to its closer proximity to open ocean regions





than the O-Buoys. The PACK and BLOW+PACK runs show the highest skill in reproducing
observations, falling within the monthly average of hourly measured $BrO_{LTcol}$ error for 9 of the 13 months
plotted in Figure 5. Both PACK and BLOW+PACK replicate the observed monthly pattern on O-Buoy 11
and at Utqiaġvik especially well. The seasonal pattern of maximum modeled $BrO_{LTcol}$ at Utqiaġvik in
March followed by a decrease to near-zero modeled $BrO_{LTcol}$ in May is replicated in both runs despite the
overprediction of $BrO_{LTcol}$ in February and March. The BLOW+PACK monthly $BrO_{LTcol}$ is between
$1*10^{14}$ molecules/cm$^2$ and $1*10^{13}$ molecules/cm$^2$ higher than PACK monthly $BrO_{LTcol}$ due to the addition
of blowing snow. This increase is most pronounced in February and March at Utqiaġvik when lower
temperatures lead to lower threshold windspeeds and increased SSA production (see Supplemental Figure
S4).

The inclusion of increased daytime yield of snowpack $Br_2$ drives monthly average $BrO_{LTcol}$ above

$3*10^{13}$ molecules/cm$^2$ in the PHOTOPACK and BLOW+PHOTOPACK runs from February until June,
far above peak observed monthly $BrO_{LTcol}$ of $2*10^{13}$ molecules/cm$^2$. The PHOTOPACK and
BLOW+PHOTOPACK runs show steady decline in $BrO_{LTcol}$ from February to June at Utqiaġvik.
Predictions of PHOTOPACK and BLOW+PHOTOPACK monthly June $BrO_{LTcol}$ above $2*10^{13}$
molecules/cm$^2$ on the O-Buoys is due to increasing photo-assisted local snowpack $Br_2$ emissions over the
Arctic Ocean (see Figure 5).  The PHOTOPACK mechanism predicts monthly average $BrO_{LTcol}$ within
observational error only on O-Buoy 12 in April. Aside from this replication of the sparsely sampled O-
Buoy 12 April $BrO_{LTcol}$, the PHOTOPACK mechanism overestimates $BrO_{LTcol}$. This overprediction of
$BrO_{LTcol}$ by PHOTOPACK and BLOW+PHOTOPACK extends to prediction of unrealistically high
mixing ratios for all tropospheric bromine species (see Supplemental Figure S7). This overprediction is a
product of high ozone deposition velocities and daytime conversion rates to $Br_2$.

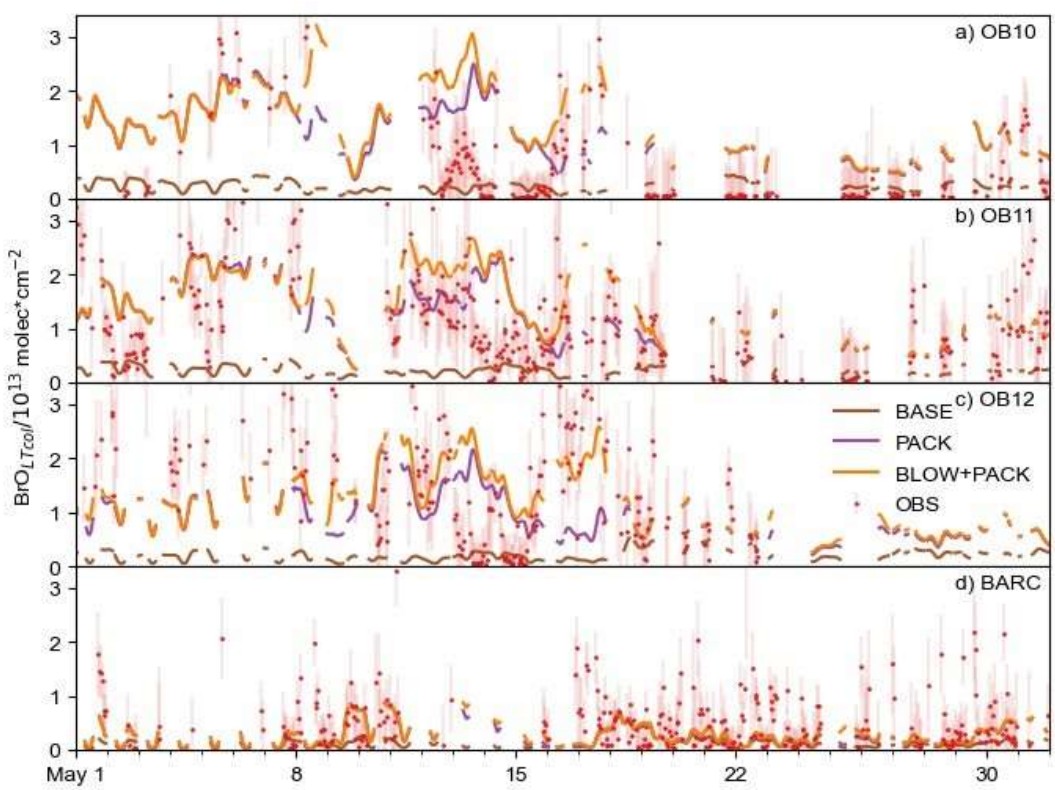


**Figure 6: Hourly BrO$_{LTcol}$ timeseries**

Hourly timeseries of BLOW+PACK, PACK, and BASE BrO$_{LTcol}$ on a) O-Buoy 10, b) O-Buoy 11, c) O-
Buoy 12 and d) BARC at Utqiaġvik in the 2015 Arctic Spring. O-Buoy observations and error bars in red,
BASE BrO$_{LTcol}$ in brown, PACK BrO$_{LTcol}$ in purple, and BLOW+PACK BrO$_{LTcol}$ in orange. All BrO$_{LTcol}$
plotted continuously except for gaps where dSCDO$_4$ >1*10$^{43}$ molecules$^2$cm$^{-5}$.

**3.4 BLOW+PACK run best replicates hourly BrO events in mid and late May**

The model's hourly predictions of BrO$_{LTcol}$ in May 2015 are shown in Figure 6 for the BASE,
PACK, and BLOW+PACK runs. The O-Buoys show fluctuations in observed BrO$_{LTcol}$ during May and
show consistent increased columns of BrO$_{LTcol}$ from May 10 to May 20. The BASE run never rises above
10$^{13}$ molecules/cm$^2$ and underpredicts most May hourly BrO$_{LTcol}$, although BASE predicts monthly
BrO$_{LTcol}$ on OB10 for two out of three months. Both PACK and BLOW+PACK runs show more skill in
replicating BrO$_{LTcol}$. The addition of the snowpack mechanism allows us to predict increased BrO$_{LTcol}$ in
late May on O-Buoys 10 and 11. This points to the role of surface snowpack in late-season events in
agreement with the findings of Burd et al. (2017).
We can identify the role of blowing snow SSA by comparing the PACK and BLOW+PACK runs.
Both PACK and BLOW+PACK runs underestimate BrO$_{LTcol}$ during the first ten days of May. BrO





predictions and observations are more active starting on May 10. The blowing snow mechanism increases
BLOW+PACK $BrO_{LTcol}$ on May 12 and 13. PACK is skilled at replicating observed O-Buoy 11 $BrO_{LTcol}$
on both days, and both PACK and BLOW+PACK are within observational $BrO_{LTcol}$ error on May 13.

A BrO event also occurs on May 13 on O-Buoy 10. While the strength of the O-Buoy 10 BrO

event is overestimated by PACK and BLOW+PACK, the shape of that event is duplicated in both runs.
Examination of the O-Buoy 10 vertical $Br_y$ profile in Supplemental Figure S7 shows surface BrO
increasing to 2 pmol/mol in the lowest 200 meters of the troposphere on May 10. BrO is mixed vertically
on May 12 and 13 throughout the lower troposphere, with a linear decrease from surface BrO mixing
ratios of 3 pmol/mol to 0 pmol/mol at 1200 m altitude. This May 12 $BrO_{LTcol}$ event is also associated with
surface ozone depletion to 15 nmol/mol.

Observed $BrO_{LTcol}$ decreases rapidly on all O-Buoys after May 14, and the model is unable to

track this sharp decrease. Rapid changes in $BrO_{LTcol}$ may be caused by sharp edges in BrO-enriched
airmasses such as those seen by Simpson et al. (2017). GEOS-Chem run at this resolution cannot replicate
abrupt changes in BrO, but it does slowly decrease $BrO_{LTcol}$ to reach $BrO_{LTcol}$ to less than $10^{13}$
molecules/cm$^2$ on May 16. The BLOW+PACK mechanism is skilled in replicating the magnitude and
features of a mid-May BrO event on several O-Buoys.

Figure 7 shows all Spring 2015 $BrO_{LTcol}$ observations on O-Buoys 10, 11, 12, and BARC plotted

against PACK $BrO_{LTcol}$ and BLOW+PACK $BrO_{LTcol}$. The increase in $BrO_{LTcol}$ on adding BLOW leads to
fewer underpredictions of observations (see bottom right section of Figure 7b). The Pearson correlation
coefficient (r) between PACK $GCBrO_{LTcol}$ and observed $BrO_{LTcol}$ is 0.33, improving to 0.39 on addition of
BLOW in the BLOW+PACK run. Other runs show less skill in replicating observations, with a BASE
$BrO_{LTcol}$ Pearson correlation to observations of 0.19 and a BLOW $BrO_{LTcol}$ Pearson correlation to
observations of 0.23. We also performed a simple linear regression to determine the relationship between
predictions and observations for each run. The slope of the line of best fit improves drastically on addition
of PACK, changing from 0.06 for BASE and 0.07 for BLOW to 0.33 for PACK and 0.44 for
BLOW+PACK. There is a positive synergistic effect on the slope of the line of best fit when using both
BLOW and PACK in combination rather than individually. The use of both BLOW and PACK
mechanisms implements literature findings on the processes influencing Arctic reactive bromine and
increases correlation between GEOS-Chem predictions and observations.

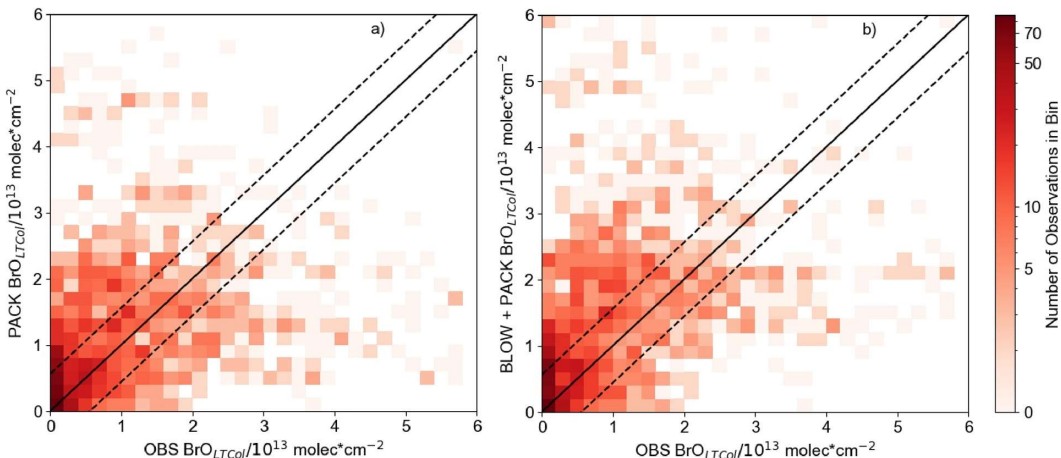


**Figure 7: Hourly modeled BrO$_{LTcol}$ versus BrO$_{LTcol}$ observations**

Two dimensional histograms showing density of GEOS-Chem predicted BrO versus all Spring 2015 hourly Br$_{LTcol}$, with a) PACK BrO$_{LTcol}$ shown at left sorted into square bins of 0.2 with an Pearson *r* correlation to observations of 0.33 and b) BLOW+PACK BrO$_{LTcol}$ on the bottom sorted into square bins of 0.2 with Pearson *r* correlation to observations to 0.39. All units are in molecules/cm$^2$. 1:1 line drawn in the center in black, with a margin of the average observational error plotted in dashed black lines around the central 1:1 line.

## 4. Arctic Spring reactive bromine modeling discussion

### 4.1 Use of both mechanisms in conjunction leads to best prediction of tropospheric BrO results

Initial implementation of this snowpack mechanism in Toyota et al. (2011) increased the daytime yield of Br$_2$ from ozone depletion to 7.5% to improve agreement between observed and modeled surface ozone mixing ratios. Toyota et al. (2011) also increased the surface resistance of ozone to 10000 s/m, decreased deposition velocities on Arctic snowpack to approximately 0.01 cm/s. Our model using a constant yield of Br from ozone deposition performs best, despite observations that sunlight has an effect on reactive bromine recycling in the snowpack (Pratt et al., 2013; Custard et al., 2017). GEOS-Chem does not explicitly model heterogeneous photochemistry within the snowpack interstitial space but does include heterogeneous bromine chemistry on aerosol particle surfaces after the Br$_2$ is emitted from the snowpack into the lowest model layer. The updates to GEOS-Chem halogen chemistry (Schmidt et al., 2016; Sherwen et al., 2016b; Chen et al., 2017; Wang et al., 2019b) should be mechanistically sufficient to model daytime heterogeneous chemistry of reactive bromine on aerosol surfaces. We note that improvements to GEOS-Chem have increased the explicit modeling of these photochemical recycling and amplification processes, possibly reducing the need for empirical increases to daytime yields.

Our findings differ from recent implementations of the snowpack mechanism in Herrmann et al. (2021) and Marelle et al. (2021). While all snowpack mechanisms are based on Toyota et al. (2011),





several large differences in model configuration and mechanism implementation explain these
differences. We allow $Br_2$ production from ozone deposition over all snow surfaces, leading to much
higher $Br_2$ production over MYI and coastal regions. Land snowpack can produce $Br_2$ on exposure to
ozone and sunlight (Pratt et al., 2013; Custard et al., 2017) and Figure 4 shows our coastal snowpack
producing large quantities of $Br_2$. Tropospheric reactive bromine chemistry has been observed up to 200
km inland from the coast (Peterson et al., 2018). Marelle et al. (2021) underestimates BrO in late March
and overestimates Utqiaġvik BrO in early April. This seasonal pattern may be due to increased daytime
ozone yield on first year ice near Utqiaġvik in April. Herrmann et al. (2021) found that HOBr and $BrNO_3$
deposition was more important in driving snowpack $Br_2$ production and that the daytime yield of 7.5%
$Br_2$ on ozone deposition underpredicted BrO. We find that ozone contributes slightly more than HOBr
and $BrNO_3$ because we allow for $Br_2$ production on ozone deposition over multi-year ice and coastal
snowpack regions. The temporal scale of this manuscript spans the entire year, while Herrmann et al.
(2021) only spans February, March, and April. Our longer timescale highlights the issue of increased
daytime $Br_2$ yield during May and June (see Figure 4 PHOTOPACK) with increased emissions over the
Arctic Ocean that are not in agreement with satellite observations of minimal Arctic tropospheric BrO in
June (Richter et al., 1998).

**4.2 Addition of PACK mechanism increases surface ozone predictive skill**

The Barrow Arctic Research Center (BARC) in Utqiaġvik has the most comprehensive coverage

of surface ozone in Spring 2015. A constant yield of 0.1% $Br_2$ from ozone deposition allows us to
approximate the average vertical extent of ozone depletion events at Utqiaġvik in May 2015. The increase
in $Br_y$ in the PACK and BLOW+PACK runs is confined to the lowest 1000 m of the atmosphere (see
Supplemental Figure S7). Ozone depletions, caused by reactive bromine chemistry, often only occur
within the lowest 1000 m of the troposphere (Bottenheim et al., 2002; Salawitch et al., 2010). Previous
studies have found evidence of lofted BrO in plumes at altitudes up to 900 m AGL (Peterson et al., 2017).
The monthly average Utqiaġvik May surface ozone in BLOW and BLOW+PACK is 22 nmol/mol,
matching mean May surface ozone from 1999-2008 (Oltmans et al., 2012). The PHOTOPACK runs
generate mean May surface ozone depletion to approximately 5 nmol/mol, far below the May mean. The
PACK and BLOW+PACK runs duplicate the approximate vertical extent of elevated bromine levels and
the strength of historic May ozone depletion.

Figure 8 shows hourly ozone predictions alongside BARC ozone observations (McClure-Begley,

Petropavlovskikh, and Oltmans, 2014). The BASE model fails to replicate variance in ozone measured at
BARC in Utqiaġvik, with a Pearson correlation coefficient to observations of 0.35. Adding PACK
improves Pearson correlation to 0.47, within rounding error of BLOW+PACK Pearson correlation of

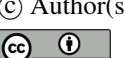

0.47. Both PACK and BLOW+PACK significantly improve model performance in replicating ozone
depletions in such as the depletion below 30 nmol/mol from March 20 to March 29 but fail to track the
subsequent recovery of ozone to background levels on April 1. Predicted PACK ozone does not recover
to backgrounds levels until a height of roughly 1000 m. A similar pattern where our model replicates low
ozone but fails to predict the recovery of ozone to background levels occurs on April 5 and 15. Previous
modeling of Utqiaġvik spring 2012 ozone in WRF-Chem found a similar linear correlation coefficient of
0.5 to BROMEX observations (Simpson et al., 2017) when using both blowing snow and snowpack
mechanisms (Marelle et al., 2021). We are biased low compared to observations, with a root mean square
error of 17.0 nmol/mol compared to a root mean square error of 12.9 nmol/mol in Marelle et al. (2021).
This may be partially due to limited vertical resolution in GEOS-Chem that may be inadequate to describe
shallow surface-based temperature inversions and subsequent recovery. The high bias in ozone deposition
velocity over sea ice surfaces may also contribute to low ozone mixing ratios near the surface.

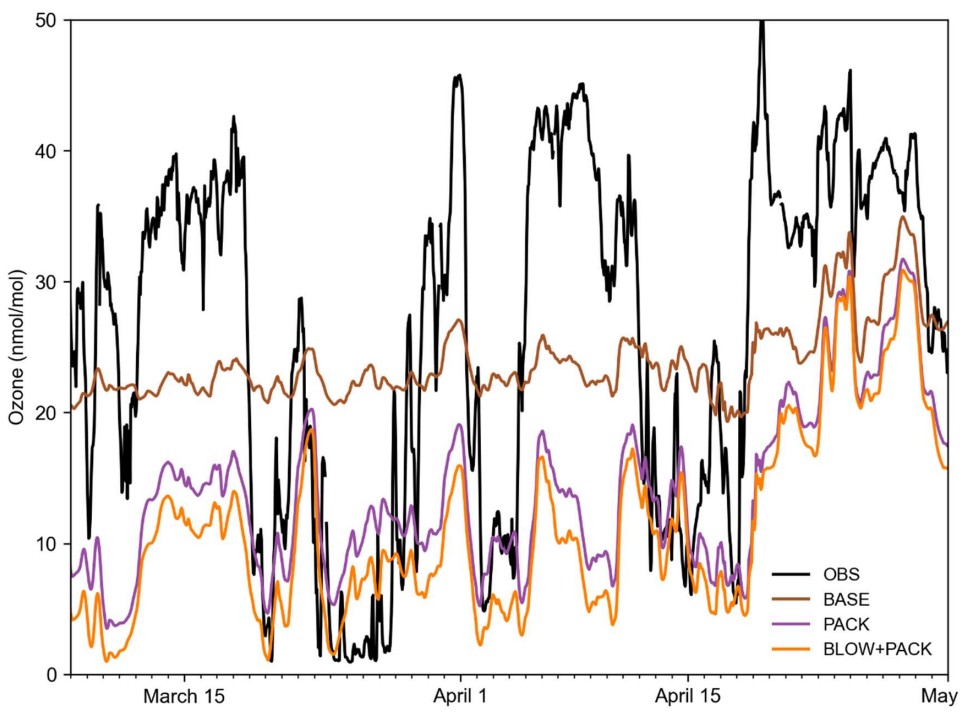


**Figure 8: Hourly Utqiaġvik ozone timeseries**

Hourly timeseries of BLOW+PACK, PACK, and BASE ozone Utqiaġvik in the 2015 Arctic Spring.
Ozone observations at BARC in black (McClure-Begley, Petropavlovskikh and Oltmans, 2014), BASE
ozone in brown, PACK ozone in purple, and BLOW+PACK ozone in orange. Gaps indicate missing
observational data.





### 5. Examining reactive bromine in the Arctic in September and October

O-Buoys deployed during fall 2015 measured BrO slant column densities characterized by noise around zero (see Supplemental Figures S8 and S9). We do not retrieve vertical column density from these fall slant column densities, because the resulting retrievals would be biased positive due to an algorithm requirement that only positive BrO column densities are allowed in the optimal estimation inversion. These differential slant column densities (dSCDs) can be used qualitatively to determine the presence or absence of BrO above the detection limit. If the dCSDs display noise around zero at all viewing angles, the BrO in the troposphere is below the detection limit of the spectrometer. The pattern of larger BrO dSCDs at near-horizon viewing elevation angles observed at Utqiaġvik during Arctic Spring in 2015 Supplemental Figure S7, indicate the presence of tropospheric BrO above the detection limit, which only occur during Arctic spring. Any BrO present in the Arctic troposphere in September and October falls below detection limits at Utqiaġvik (see Supplemental Figure S8) and on each O-Buoy (see Supplemental Figure S9). The average Arctic Spring 2015 MAX-DOAS $BrO_{LTcol}$ detection limits are 5 x $10^{12}$ molecules/cm$^2$ (Peterson et al., 2015; Simpson et al., 2017; Swanson et al., 2020). Both BLOW and PACK mechanisms lead to prediction of increased fall BrO because the weather and sea ice conditions specified in the emission algorithms occur in fall as well as spring.

Figure 9 shows fall predictions of $BrO_{LTcol}$ filtered for times when solar elevation angle was greater than 5°. BASE and SNOW $BrO_{LTcol}$ remain near zero in September but rise above the MAX-DOAS detection limit of 5 x$10^{12}$ molecules/cm$^2$ $BrO_{LTcol}$ in October. The addition of the blowing snow mechanism propels BLOW $BrO_{LTcol}$ up to 6 x $10^{13}$ molecules/cm$^2$ in October. O-Buoys 13 and 14 have the highest modeled fall $BrO_{LTcol}$ but even Utqiaġvik has several days of $BrO_{LTcol}$ above 5 x$10^{12}$ molecules/cm$^2$ in late October. There is no clear evidence of any BrO above MAX-DOAS detection limits at Utqiaġvik or on any O-Buoy in October, as seen by the dSCDs scattered around zero in Supplemental Figures S8 and S9.



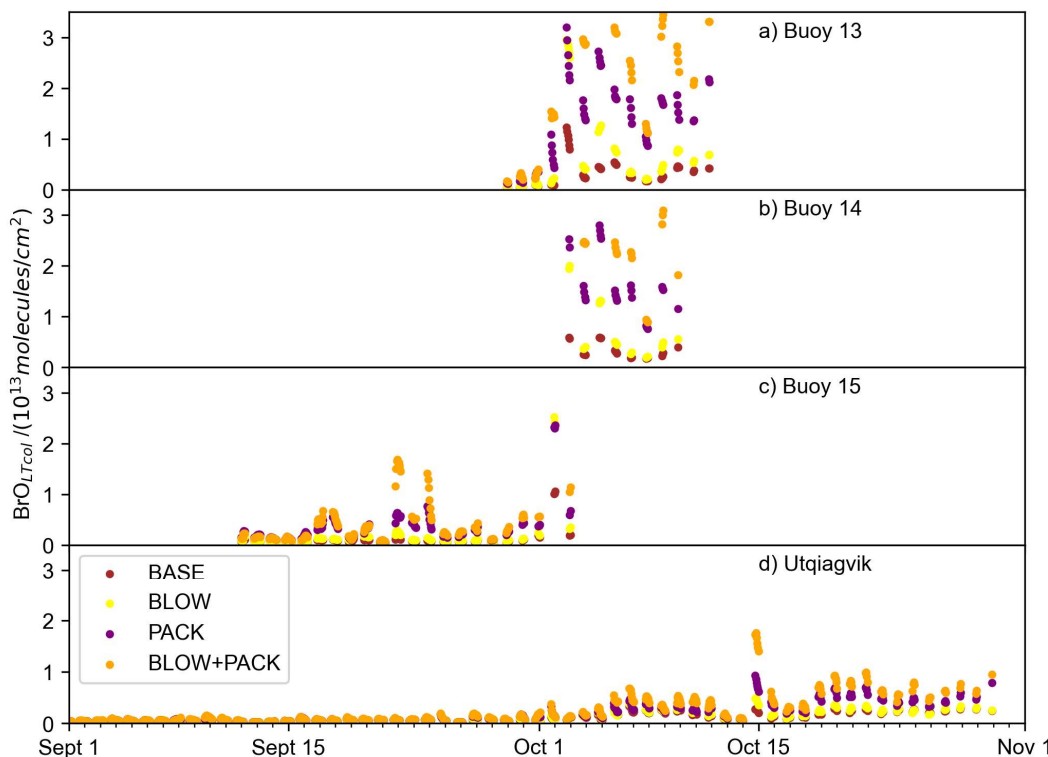

**Figure 9: Fall GEOS-Chem Predicted BrO$_{LTcol}$**

Hourly timeseries of BLOW+PACK, PACK, and BASE BrO$_{LTcol}$ on a) O-Buoy 10, b) O-Buoy 11, c) O-Buoy 12 and d) BARC at Utqiaġvik during September and October 2015 BASE BrO$_{LTcol}$ in brown, PACK BrO$_{LTcol}$ in purple, and BLOW+PACK BrO$_{LTcol}$ in orange. All BrO$_{LTcol}$ plotted continuously except for gaps where solar elevation angle was less than 5˚.

Both mechanisms assume that snowpack and SSA are just as capable of recycling reactive bromine as in the springtime. High fall and winter SSA agrees with observations of peak SSA during polar winter in both Antarctica (Wagenbach et al., 1998) and in the Arctic (Jacobi et al., 2012). The deposition of Arctic haze (Douglas and Sturm, 2004) and SSA (Jacobi et al., 2019) increases snowpack salinity and sulfate content over the course of winter and spring. This seasonal change in snowpack salinity and acidity may enable reactive bromine recycling in the Arctic Spring, but there may not sufficient haze and SSA deposition in fall to decrease snowpack pH and increase snowpack bromide content. Additional observations of fall snowpack over sea ice including ion content could show different snowpack composition in spring and fall. Thus the GEOS-Chem model overestimates fall BrO by assuming the fall snowpack is equally capable of reactive bromine recycling as spring snowpack, possibly due to the assumption of an infinite reservoir of snowpack bromide in all seasons. Most other modeling



exercises have focused on spring with unknown predictions in fall, possibly indicating problems in
mechanisms or parameterizations being employed, so we suggest that modeling should be done for a full
year to improve underlying chemistry and physics.
**6. Conclusions**

We add snowpack $Br_2$ production to GEOS-Chem based on multiple field observations

demonstrating molecular bromine production in snowpack interstitial air. We use a mechanistic
parameterization of snowpack $Br_2$ production based on Toyota et al. (2011) in which $Br_2$ is emitted from
all snowpack of sufficient salinity and depth over land and sea ice upon deposition of the precursor
species HOBr, $BrNO_3$, and ozone. Prior work has also added a blowing snow SSA production mechanism
that increases aerosol particulate bromide and thus facilitates heterogeneous recycling of reactive bromine
on these aerosol particle surfaces. We update the halogen scheme to GEOS-Chem 12.9.3 and performed
six model simulations including a BASE run with neither blowing snow nor snowpack emissions, a
PACK run assuming constant yield of $Br_2$ on ozone deposition over all snow surfaces, a PHOTOPACK
run assuming increased daytime yield of $Br_2$ on ozone deposition (similar in Toyota et al., 2011), a
BLOW run using only blowing snow SSA formation and two additional runs combing BLOW and each
respective PACK mechanism. The increased daytime yield of $Br_2$ in PHOTOPACK leads to
overprediction of BrO in these simulations, but the PACK run (with constant $Br_2$ yield day and night)
matches monthly averaged BrO vertical column densities for 9 of 13 cases at O-Buoy and Utqiagvik in
springtime months. The PACK and BLOW+PACK runs were successful in replicating observed mid-May
BrO events on O-Buoys as well as recurrence events at the end of May. The BLOW mechanism
effectively increases aerosol surface available for turnover of reactive bromine. The snowpack
mechanism has more impact on modeled BrO mixing ratios than the blowing snow mechanism, but both
contribute to tropospheric reactive bromine. We extend our model run to the full year and find that
enhanced daytime $Br_2$ yield can lead to increased Arctic Ocean $Br_2$ production in the summer. Examining
modeled BrO in fall 2015 reveals prediction of BrO when using these mechanisms that are at odds with
observations.

The inclusion of two Arctic reactive bromine production mechanisms based on literature

observations of snowpack $Br_2$ emission and blowing snow SSA formation improves model skill in
replicating Arctic tropospheric BrO in spring 2015. The snowpack is an important source of reactive
bromine, and SSA particles provide an abundant surface for sustained reactive bromine recycling in the
troposphere. We find that using both snowpack and blowing snow bromine production mechanisms is
necessary for modeling BrO in the Arctic.
*Competing interests:* The authors declare that they have no conflict of interest.



Author contributions. WFS, WRS and CH designed the study. WRS collected and curated MAX-DOAS data. KC, LM, JT, LJ, JH and contributed code for reactive bromine mechanisms. CH, KC, LJ, JH, BA, SZ, QC, XW, and TS contributed model updates. WFS carried out modeling and analysis. WFS wrote the paper with input from all authors.

## 7. Acknowledgements

We acknowledge support from the National Science Foundation for providing funding under grants ARC-1602716, AGS-1702266, AGS-2109323, and ARC-1602883. This work also supported by the CNRS INSU LEFE-CHAT program under the grant Brom-Arc, and NASA grant 80NSSC19K1273. This research has received funding from the European Union's Horizon 2020 research and innovation program under grant agreement no. 689443 via project iCUPE (Integrative and Comprehensive Understanding on Polar Environments). The O-Buoy and Utqiagvik ground-based BrO datasets are available in the arcticdata.io repository (doi:10.18739/A2WD4W). We recognize the work of Jiayue Huang in adding the blowing snow SSA mechanism to GEOS-Chem. We would like to thank the National Oceanic and Atmospheric Administration (NOAA) Global Monitoring Division for the provision of ozone and temperature data near Utqiaġvik available online at doi:10.7289/V57P8WBF. We acknowledge use of the coastline distance dataset from the Pacific Islands Ocean Observing System. We acknowledge the use of imagery from the Land Atmosphere Near Real-Time Capability for EOS (LANCE) system and services from the Global Imagery Browse Services (GIBS), both operated by the NASA/GSFC/Earth Science Data and Information System (ESDIS, https://earthdata.nasa.gov) with funding provided by NASA/HQ. We owe a debt of gratitude to all members of the Atmospheric Chemistry and Global Change group at Florida State University for their support for working with GEOS-Chem and Python. We thank the global GEOS-Chem community for their tireless work to improve the model. We also thank all involved in the O-Buoy project for data collection and analysis.

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
