# Peer review of "Comparison of model and ground observations finds snowpack and blowing snow aerosols both"

_Atmospheric Chemistry and Physics, 2022_

## Author Response (AR1)

Responses to Reviewer 1

Specific comments:

Does the self-reaction of BrO only produce $Br_2$ (via R6) in the gas-phase chemical mechanism of GEOS-Chem? $Br_2$ should rapidly dissociate to Br-atoms by sunlight anyway, but I think a majority of Br-atoms are in fact produced directly via BrO + BrO (see https://iupac-aeris.ipsl.fr/htdocs/datasheets/pdf/iBrOx22_BrO_BrO.pdf). If this is true in the chemical mechanism of GEOS-Chem as well, the authors need to adjust their statement in the second paragraph of introduction, the diagram in Figure 1 and possibly the calculation for R1 presented in Table S1.

Self reaction of BrO also produces two Br atoms in GEOS-Chem (Parrella et al., 2012). The photolysis of Br2 occurs readily to form two Br atoms, and we have lumped these two rates together for our calculation of R6 in Table S1. We have clarified this in the introduction paragraph and the caption of Figure 1. R6 increases in importance in our model run on the addition of PACK, making BrO self-reactions for those model runs one of the most rapid reactions outside of the reactions involved BrOx cycling.

On Lines 78-79, the authors state: "… bromine nitrate ($BrNO_3$), which can undergo hydrolysis on a saline surface to form HOBr as in HR3". The reaction of $BrNO_3$ on the "saline surface" may rather end up in the formation of BrCl and $Br_2$. It is probably more accurate to say, the hydrolysis of $BrNO_3$ occurs on aqueous and ice surfaces to form HOBr.

We have updated this sentence to your recommended wording.

In the caption of Figure 1 (Line 84), the authors state: "all gaseous species may undergo wet and dry deposition". Does it mean that even Br and BrO undergo wet and dry deposition in GEOS-Chem?

I have updated this sentence to reflect that all gaseous species can undergo dry deposition. Wet deposition of p-Br is shown in the figure.

Lines 276-277: How are these five particles per snow grain distributed between difference size bins of aerosols? Also, is there not any dependence on ambient relative humidity assumed in the production of SSAs from each snow grain?

The size distribution of the aerosols created by the BLOW mechanism follows a two-parameter gamma distribution following Yang et al. (2008). The value of five particles per snow grain was chosen based both on sensitivity studies and similarity to particle size distributions observed at Utqiaġvik (Huang and Jaeglé, 2017). The dependence on ambient RH is encompassed by the threshold requiring relative humidity to be below saturation with respect to ice.

Section 2.5: It is not very clear what is assumed for the role of temperature in the snowpack emissions of molecular bromine. Does the model assume that the $Br_2$ emissions can occur in the same way as long as temperature is lower than 0 degree Celsius? Also, is the seasonal (summer) snowmelt not assumed to deactivate the capability of the snowpack for producing $Br_2$ as discussed by Burd et al. (2017)?

We assume in this simplified model that emissions can occur in the same way as long as the temperature is lower than 0 °C. It would be possible to halt all future production of Br2 from snowpack after temperature rises above 0 °C (assuming instant snowmelt). While Burd et al. (2017) defined the melt onset date as the first time the 3-hour average temperature reached 0 °C. However, there can be a resumption of reactive bromine chemistry following the melt onset date ( called a recurrence event in Burd et al. (2017)) and we wanted to include the potential for these events to occur. We also felt that choosing a duration above 0 °C after which all surface snowpack has melted would be arbitrary.

Line 364: Bariteau et al. (2010) estimated the dry deposition velocities of ozone on open oceans from shipboard ozone flux measurements. How has this information been translated to the parameterized dry deposition velocities of ozone on sea ice in GEOSChem?

Pound et al. (2020) updated GEOS-Chem to use a representation of oceanic ozone deposition based on its reaction with sea-surface iodide (Sherwen et al., 2019). This update was included in GEOS-Chem 12.8 and our model runs. However, this update to the ozone deposition scheme does not affect ozone deposition over sea ice within the model.

Line 468-469: It appears to me from Figure 4 that dry deposition velocities are higher (instead of lower) over the ice-covered ocean than over the open ocean in GEOS-Chem. The authors may want to reconsider the argument here and in the subsequent sentence.

I have removed this sentence, as the source does not support this argument. The Toyota 2016 paper only does direct comparisons between bulk Richardson numbers over sea ice and open ocean, finding that the air over sea ice is more likely to be stable. While they do discuss deposition velocities, their discussion is in terms of the sensitivity of ozone deposition velocity over ice to the choice of stability correction algorithm, and does not make any blanket statements about the relative deposition velocities over sea ice and open ocean. Their discussion of deposition velocity calculation is of interest for deposition scheme design but not relevant to this paper.

L538-539: The authors should elaborate on what they believe are the realistic ranges of mixing ratios for "all tropospheric bromine species" based on some observed values reported in the past.

This comparison has been eliminated as we feel that a direct comparison between in-situ sampled observations of BrO mixing ratios and large grid average layer compositions are inappropriate in this context. We believe PHOTOPACK produces a large overestimation of monthly BrY but the bulk of this overestimation comes within the large p-Br mixing ratios near the surface. We have gathered data on BrO mixing ratios at Utqiagvik for five years and the PHOTOPACK runs predict roughly double the average observed spring Utqiagivik BrO surface mixing ratios, which is an overprediction but is believable given the interannual variability in BrO (Swanson et al., 2020). Our instruments did not gather aerosol bromide content data and we cannot prove that PHOTOPACK p-Br$^-$ predictions are outside of a realistic range.

I frequently encountered unclear or inconsistent statements in Section 3.4. I recommend the authors to proofread again. Here are the problems I have noticed: "BASE predicts monthly BrO$_{LTcol}$ on OB10 for two out of three months" – be more specific (Line 551-552),

I have listed the months in which it was most accurate.

"BrO predictions and observations are more active starting on
May 10" – clarify what the "more active" means (Lines 557-558),

I have specified that this refers to the increase in predicted BrO variability and higher BrO peaks.

Figure S7 does not
seem to be showing the Bry profiles over O-Buoy 10 (Lines 563-564),

I have eliminated this paragraph, which referred to figures originally found in my thesis presentation that are not included in this manuscript.

and "The
BLOW+PACK mechanism is skilled in replicating the magnitude and features of a mid-May BrO event on several O-Buoys" – clarify what the "several" indicates (Lines 572-573).

I have specified O-Buoys 10 and 11 here.

Section 4.2: Apparently, even the BASE model under-predicts the background mixing ratios of surface ozone by more than 10 ppb. It is probably worth mentioning the northern hemispheric tropospheric ozone bias in the current generation of GEOS-Chem as indicated in Wang et al. (2021,https://acp.copernicus.org/articles/21/13973/2021/).

I have added in a citation of the Wang paper and its finding that tropospheric ozone is underpredicted at high latitudes after the sentence in line 649.

Line 724: The authors should elaborate on what "recurrence events" mean exactly and why they are important or interesting, perhaps in earlier sections.

Burd et al. (2017) defines recurrence events (an increase in BrO after the termination date) in relation to termination date (date at which BrO has been within error of 0 for five days). As we do not have enough continuous May 2015 data on the O-Buoys to define a termination event, we have eliminated this vocabulary and instead noted that GEOS-Chem does a good job predicting BrO events on all O-Buoys in May 2015.

Line 724-725: The authors should discuss the enhancement of aerosol surface areas arising from SSAs in the BLOW mechanism in a quantitative manner (e.g., as compared to the surface areas of more persistent springtime haze aerosol particles), in earlier sections where the difference in model behavior between the PACK and BLOW+PACK cases is discussed.

We have rephrased this sentence to discuss particulate bromide availability. We briefly discuss the increase in particulate bromide leading to increased BrO in section 3.3. As the particle size distribution produced from BLOW remains constant, any change in surface area is in fact related to the total mass produced, and therefore discussing particulate bromide lends more clarity to our argument.

Figure S6: If I understand correctly, the clear-sky screen is performed by rejecting the instances with the lofted layer degrees of freedom (lofted layer DOF) lower than 0.5,

which in general corresponds to cloudy instances with dSCDs of $O_4$ greater than $1.0 \times 10^{43}$ molecules$^2$ cm$^{-5}$. However, in the plot shown, I do not see the increase of the lofted layer DOF with decreasing dSCDs of $O_4$ (clearer sky), but actually the increase of the lofted layer DOF with increasing dSCDs of $O_4$ (cloudier sky). Do I misunderstand something here?

      While the relationship is complicated by the differential nature of our measurements, clear skies have higher $O_4$ dSCDs than cloudy skies. We mention on line 443 that clear sky conditions have higher dSCD $O_4$. The slant column density of $O_4$ is proportional to the path the incident light travels through the atmosphere. A longer path means more absorption by $O_4$, and clouds and aerosols effectively shorten the path length.

Figure S7: The $O_3$ profiles are not shown for the BASE, BLOW, PACK and BLOW+PACK model results. It will be useful to include them as well.

      With multiple small figures, we wanted to retain clarity in each subplot. The X-axis scale on the PHOTOPACK figures was such that we could plot ozone in nmol/mol and Br-eq in pmol/mol without distorting the graph. To add ozone to the other four graphs, we would have to add an additional set of tickmarks on top of an already crowded graph. We have made and examined these graphs and found that ozone is somewhat depleted at the surface but not nearly to the extent that it is depleted in the PHOTOPACK runs. I have attached Test Figure 1 showing the vertical extent of ozone depletion in May for the PACK run. While this paper does not focus on the ozone depletion aspect, we realize that there may be readers interested in a more detailed look at ozone depletions. We have prepared animations of the hourly vertical ozone profile at each observation location for May 2015 for each model run and are happy to share them on request.

[Figure]

Test Figure 1 Details: Average May vertical ozone profile for the PACK run. Ozone in pmol/mol is depicted on horizontal axis, height above ground in km depicted on vertical axis.

Table S1: As indicated in the specific comment #1, I wonder if the self-reaction BrO + BrO can directly produce Br-atoms in the GEOS-Chem gas-phase chemical mechanism. If it does, R1 should include its contribution.

      This reaction has been lumped into R6, as both involve the reaction of two BrO molecules.

Table S3 contains some important values that should be clearly mentioned and

discussed in the text. Comparison between the values of "Emission PACK Br$_2$" and "Emission SSA p-Br$_-$" should be referred to when discussing their respective impacts on the simulated concentrations of gaseous bromine species in the model. I assume that the "Emission SSA p-Br$_-$" include contributions from both bubble bursting on open oceans and the production of SSAs from blowing snow. Please correct if I am wrong. I would also like to confirm that the changes in "Emission SSA p-Br$_-$" are indeed less than 1 millions of moles per hour across the region by switching on the BLOW mechanism in the model.

The emitted SSA p-Br- quantity does include both bubble bursting and SSA production from blowing snow. The change in this quantity from adding blowing snow is an increase of 390,000 moles per hour on adding BLOW to BASE, but a smaller decrease of 210,000 moles per hour is seen on adding BLOW to PACK. The values in this table are somewhat misleading, as the particulate bromide is not equal to the total particulate mass emitted, nor is it equal to the amount of reactive bromine added to the troposphere. This table provides raw numbers on emission of bromine across a specific region, but we do not feel that a table is the best way to convey this information. We feel that Figure 4 better conveys the spatial differences and patterns in emission.

Tables S2 and S3 were included as a reference for other modelers who may be interested in the magnitude of dry deposition terms in our model. Neither are mentioned in the main text and we have decided to eliminate them from the supplement. We have moved table S1 to the main text to provide supporting information for Figure 1 and provide specific details on reaction rates within GEOS-Chem. We are happy to provide these tables and any other bromine model budgets to interested parties.

Specific notes:
We have made all grammatical and technical changes in accordance with reviewer recommendations.

L444-445: From the sentence, I presume that the authors have computed differential slant column densities of O$_4$ with 3-D meteorological fields of GEOS-Chem (or MERRA-2). Which variables are used in this computation?

We did not compute dSCDs of O$_4$ within GEOS-Chem. We have updated this sentence to reflect that fact that the clear sky screen was only applied to observations, which were then compared to modeled BrO quantities.

Papers cited in responses:

Burd, J. A., Peterson, P. K., Nghiem, S. V., Perovich, D. K. and Simpson, W. R.: Snow Melt Onset Hinders Bromine Monoxide Heterogeneous Recycling in the Arctic, J. Geophys. Res. Atmos., 1–13, doi:10.1002/2017JD026906, 2017.

Huang, J. and Jaeglé, L.: Wintertime enhancements of sea salt aerosol in polar regions consistent with a sea-ice source from blowing snow, Atmos. Chem. Phys., (November), 1–23, doi:10.5194/acp-2016-972, 2017.

Parrella, J. P., Jacob, D. J., Liang, Q., Zhang, Y., Mickley, L. J., Miller, B., Evans, M. J., Yang, X., Pyle, J. A., Theys, N. and Van Roozendael, M.: Tropospheric bromine chemistry: Implications for present and pre-industrial ozone and mercury, Atmos. Chem. Phys., 12(15), 6723–6740, doi:10.5194/acp-12-6723-2012,

2012.

Sherwen, T., Chance, R. J., Tinel, L., Ellis, D., Evans, M. J. and Carpenter, L. J.: A machine learning based global sea-surface iodide distribution, , (March), 1–40, 2019.

Swanson, W. F., Graham, K. A., Halfacre, J. W., Holmes, C. D., Shepson, P. B. and Simpson, W. R.: Arctic Reactive Bromine Events Occur in Two Distinct Sets of Environmental Conditions : A Statistical Analysis of 6 Years of Observations Journal of Geophysical Research : Atmospheres, , 1–19, doi:10.1029/2019JD032139, 2020.

Yang, X., Pyle, J. A. and Cox, R. A.: Sea salt aerosol production and bromine release: Role of snow on sea ice, Geophys. Res. Lett., 35(16), 1–5, doi:10.1029/2008GL034536, 2008.

Response to Reviewer 2

SPECIFIC COMMENTS
l21 reactions on wind blown snow and on aerosol (e.g. Hara et al., 2018)

We have added a mention of aerosols as possible reaction source and cited Hara et al. (2018). Furthermore, we have standardized our language throughout the paper and in our title to specify that we use a blowing snow SSA mechanism rather than a blowing snow mechanism, as Hara et al. (2018) and others make a convincing case that the blowing snow itself is not the primary source of increased reactive bromine.

l33-34 This is somehow not complete as SSA from blowing snow does not only provide additional surface area for heterogeneous halogen chemistry but also additional bromide. Please clarify and add detail how aerosol is treated in the model as a finite bromide reservoir. How does modelled bromide depletion in aerosol compare to observations?

I have clarified in the introduction to specify that the bromide on those aerosols is the direct source of increased BrO. We follow the same bromide treatment detailed in Huang et al. (2020), where they examine not only the depletion of aerosol but the deposition of bromide onto surface snow. They found that their predicted surface snow enrichment was consistent with observations of atmospheric chemical deposition to surface snow at Alert from Macdonald et al. (2017).

l34-36 This statement refers to the model calculation, which cannot be corroborated for the entire region by comparison with the observations used in this study (see above general comment). Please clarify.

These lines are only intended to report on model predictions driven by the snowpack and blowing snow sources. I have specified that these lines refer only to modeled BrO.

l97 how is pH of the heterogeneous phase (snow, aerosol) treated in the model?

Aerosol is emitted within GEOS-Chem as alkaline based on the salinity of seawater. When this alkalinity is titrated out, the aerosol is considered to have a pH of 5. I included a description of the alkalinity titration in Section 2.3.

l201-202 Is it reasonable to assume the bottom 200m are well mixed during the period

considered considering that surface inversions occur typically in winter/early spring? Please expand.

       The $BrO_{pptv200}$ quantity derived by assuming that the lowest 200 m are well mixed is not used in this paper. This quantity is included in our datasets, but as its derivation is described in the metadata we have eliminated these unnecessary lines. We are aware that is not reasonable to assume that the entire lowest 200 m of the Arctic troposphere are well mixed; for example, the OASIS field campaign measured mixing layer heights of 50 m during ozone depletion events (Boylan et al., 2014). The $BrO_{pptv200}$ quantity is of some use as a rough comparison between predicted surface mixing ratios and MAX-DOAS measurements, but the method for direct comparison detailed in Section 2.7 should be used for more rigorous model comparison to MAX-DOAS measurements.

l220-21 Heterogeneous chemical reactions can convert SSA-transported bromide into gaseous reactive bromine species in the atmosphere. How is this modelled? What pH is assumed/calculated for open ocean and blowing snow SSA?

       This is modeled as a heterogenous reaction on the aerosol surface. The pH of the SSA is based on the composition of seawater, which grants the aerosol a specific alkalinity. This alkalinity can be titrated within GEOS-Chem by deposition of acidic gases to a minimum of pH=5 (Alexander et al., 2005).

l232--233 even if all thresholds are met there needs to be snow on sea ice present to get airborne, thus this is a potential SSA production rate and therefore an upper limit. Please clarify.

       We clarify our previously unstated assumption that snowpack exists on all sea ice surfaces during the Arctic Spring. We believe that this is a reasonable assumption even on first-year sea ice freshly formed in fall due to the accumulation of snow over the winter months.

l245 great to use high resolution wind speed data (0.5x0.625º) to capture wind gusts, relevant to both BSn and open ocean SSA!

       We find these high-resolution SSA datasets very useful for reducing model runtime and generating resolution-independent results. These results may not be appropriate for all applications because they rely on specific constant values chosen in the blowing snow SSA mechanism (see Section 2.4).

l266--272 rather than choosing a single value based on a few point measurements it seems more sensible to explore the sensitivity of this important parameter with a few sensitivity model runs; same applies for SSA/SP ratio (l277) and the Br- enrichment factor

       The computational expense of running a full year at high resolution is high, and follow the findings of Huang et al. (2020) in which the performed several sensitivity studies over in March and April and settled on the parameters used here based on comparisons to satellite BrO observations.

l345-47 how did you decide on 10cm snow depth threshold? shallow snowpacks near the coast may contain enough Br- for significant halogen activation; how much area is affected by this filter?

While snow with a depth of less than 10 cm may be able to recycle snow, it will be less effective at producing reactive bromine that deeper snow. We decided on the 10 cm snow depth threshold based on detailed snowpack modeling studies, especially Thomas et al. (2011) predicting active bromine chemistry in interstitial air at 10 cm or below. The 10 cm snow depth screen mainly affects grid cells in more southerly latitudes and has little effect on Arctic production. Test Figure 2 shows the cells with snow affected by the 10 cm depth screen on May 1 2015. The perimeter of colored grid cells with snow depth less than 10 cm is further south earlier in the year due to cold winter temperatures.

[Figure]

Test Figure 2 Details: Snow depth in MERRA-2. Grid cells shown in bright yellow have snow depth greater than 0.1 m, cells shown in pink and orange represent MERRA-2 grid cells with a snow depth greater than zero and less than 0.1 m that are eliminated by our screen. White grid cells have zero land snow depth in MERRA-2 files.

l396-97 Please elaborate how the calculation of cloud pH is improved. Given that the multiphase reaction of bromide to reactive bromine depends on acidity explain also how snow pH and aerosol pH are computed or assumed in the model (see previous comment).

We have specified how SSA pH is dependent on titration of aerosol alkalinity in Section 2.3. Shah et al. (2020) increased minimum background HCOOH, performed sensitivity analysis, and adjusted key parameters such as aerosol mass partitioning, gas-water equilibration time to improve the accuracy of GEOS-Chem cloud pH calculation. We would like to direct readers to Shah et al. (2020) for more detail on adjustments to cloud pH calculations within GEOS-Chem.

l470-71 this sentence is in contradiction to the previous that O3 dry deposition should be lower above ice covered ocean than above open ocean. Please clarify.

The Toyota 2016 paper only does direct comparisons between bulk Richardson numbers over sea ice and open ocean, finding that the air over sea ice is more likely to be stable. While they do discuss deposition velocities, their discussion is in terms of the sensitivity of ozone deposition velocity over ice to the choice of stability correction algorithm and does not make any blanket statements about the relative deposition velocities over sea ice and open ocean.
I have removed the first sentence, as their discussion of deposition velocity calculation is of interest for deposition scheme design but not relevant to this paper.

l552-53 Please include a table and list a quantitative measure of model skill (e.g. root mean square error). I am not sure why only May is discussed here, I would like to see also the other months. Thus do a month-wise comparison between observations and model using the hourly data Feb-June in Table and include figures as Fig6 also for each month, possibly in the appendix.

We have calculated RMSE for BrO on PACK, BLOW, BLOW+PACK, PHOTOPACK, and BLOW+PHOTOPACK in Table 2. We discuss these model skill measurements in Section 4.1.
We choose to highlight May here as it has the best data coverage for visual inspection. Our O-Buoy observations start on April 22 and end by June 10, an additional 17 days outside of May. A discussion of monthly O-Buoy accuracy would be skewed by number of valid observations. We have included graphs of hourly BrO predictions vs observations for the entire spring season as supplemental figures S7 and S8.

Section 4.1 Overall it appears the Br2 yield from snowpacks is limited by surface resistance and ozone deposition and not availability of sunlight. Can you comment?

We note that ozone deposition makes up over half of our $Br_2$ yield. Availability of sunlight is not generally an issue, as the sun is up for the majority of the day over the Arctic Ocean after the vernal equinox. The sunlight scheme does increase the ozone yield 75-fold, so while sunlight is not the limiting factor it plays an important role in the PHOTOPACK runs.

l628 Section 4.2 It would be very informative and strengthen the paper if the modelled O3 was compared also to O-Buoys data. O3 is measured by the O-Buoy platforms, so are data not available for the time periods considered?

O-Buoy 10 did not collect any ozone data in 2015, and O-Buoy 12 did not have any overlapping ozone and BrO observations. The ozone data that was gathered by O-Buoys in Spring 2015 has a number of data gaps, and is more difficult to analyze than the reliable BARC station ozone data. I have included graphs of ozone on O-Buoys 11 and 12 in the supplement as Figures S10 and S11.

l706 not only modeling should be done but also snow sampling and analysis as surely there are no or few data to back up your speculations.

We have been told by experimental groups that fall sampling of snowpack is difficult due to the fractious nature of the ice at this time, making snow sampling methodology difficult. We have included a call for fall sampling and analysis in the manuscript to encourage such an effort.

l722-23 match within the uncertainty? This is not very quantitative, use a quantitative measure of model skill throughout as suggested above.

Our results shown here match within measurement error. Table 2 calculates model RMSE for O-Buoys 10, 11, 12 and at Utqiagvik and discusses these numbers in Section 4.1.

l727 "We extend our model run to the full year and find that enhanced daytime Br2 yield can lead to increased Arctic Ocean Br2 production in the summer" But this in disagreement with observations, as there are no ODEs in summer, please explain.

We note in Section 3.1 lines 474-475 that satellites see minimum BrO over that Arctic ocean in summer, which is at odds with the model predictions using an enhanced daytime $Br_2$ yield. We agree with your statement, and we hope that modelers reading this paper will consider this when deciding upon their snowpack $Br_2$ production schemes.

Fig1/l70-71: I find the figure confusing, at least use colour for heterogeneous reaction arrows (instead of dashed line) and associated text. I'd also suggest to include the Supplemental Table S1 into the main text to better follow the discussion and have a reference for each reaction right next to Fig1.

We have moved Supplemental Table S1 to Section 4 in the main text as Table 2 for clarity. We have referenced this table in the caption for Figure 1.

Fig2, check caption: MODIS image is the main figure, inset map shows the image footprint

We have corrected this caption.

FigS4 - show also for O-Buoy position (extracted from the model run) and discuss

We have used the highly reliable data from sensors at BARC for this graph. The O-Buoy windspeed sensors have data gaps due freezing over, and we feel that the high-quality BARC sensor data does the best job at illustrating the periodic nature of BLOW SSA production.

TECHNICAL CORRECTIONS
l224-25 grammar. drop "which"?
    Done

l450-51 is this part of Fig4 caption? please remove separating line (similar for most captions)
    Changed caption formatting throughout the paper
l598 ozone deposition?

    Updated to deposition from depletion.

l680 PACK?? please check

Good catch, old SNOW run name has been replaced with current PACK term.

all FIGURES' reproduction is fuzzy, use vector graphic.

I have prepared this manuscript in Word. I have prepared most of these figures at 300 dp, but I believe that word compresses them when converting to PDF. I have replaced several fuzzy figures with cleaner figures, and I will contact the editor to supply them with original images for final publication as needed.

Papers cited in response to reviewer:

Alexander, B., Park, R. J., Jacob, D. J., Li, Q. B., Yantosca, R. M., Savarino, J., Lee, C. C. W. and Thiemens, M. H.: Sulfate formation in sea-salt aerosols: Constraints from oxygen isotopes, J. Geophys. Res. D Atmos., 110(10), 1–12, doi:10.1029/2004JD005659, 2005.

Boylan, P., Helmig, D., Staebler, R., Turnipseed, A., Fairall, C. and Neff, W.: Boundary layer dynamics during the Ocean-Atmosphere-Sea-Ice-Snow (OASIS) 2009 experiment at Barrow, AK, J. Geophys. Res. Atmos., 119(5), 2261–2278, doi:10.1002/2013JD020299, 2014.

Hara, K., Osada, K., Yabuki, M., Takashima, H., Theys, N. and Yamanouchi, T.: Important contributions of sea-salt aerosols to atmospheric bromine cycle in the Antarctic coasts, Sci. Rep., 8(1), doi:10.1038/s41598-018-32287-4, 2018.

Huang, J., Jaeglé, L., Chen, Q., Alexander, B., Sherwen, T., Evans, M., Theys, N. and Choi, S.: Evaluating the impact of blowing snow sea salt aerosol on springtime BrO and O3 in the Arctic, Atmos. Chem. Phys., 1–36, doi:10.5194/acp-2019-1094, 2020.

MacDonald, K. M., Sharma, S., Toom, D., Chivulescu, A., Hanna, S., Bertram, A. K., Platt, A., Elsasser, M., Huang, L., Tarasick, D., Chellman, N., McConnell, J. R., Bozem, H., Kunkel, D., Duan Lei, Y., Evans, G. J. and Abbatt, J. P. D.: Observations of atmospheric chemical deposition to high Arctic snow, Atmos. Chem. Phys., 17(9), 5775–5788, doi:10.5194/acp-17-5775-2017, 2017.

Shah, V., Jacob, D. J., Moch, J. M., Wang, X. and Zhai, S.: Global modeling of cloud water acidity , precipitation acidity , and acid inputs to ecosystems, Atmos. Chem. Phys, 12223–12245, 2020.

Thomas, J. L., Stutz, J., Lefer, B., Huey, L. G., Toyota, K., Dibb, J. E. and Von Glasow, R.: Modeling chemistry in and above snow at Summit, Greenland - Part 1: Model description and results, Atmos. Chem. Phys., 11(10), 4899–4914, doi:10.5194/acp-11-4899-2011, 2011.